# Identification of undetected SARS-CoV-2 infections by clustering of Nucleocapsid antibody trajectories

Leslie R. Zwerwer [1,2,3] ✉, Tim E. A. Peto[1,4,5,6,24], Koen B. Pouwels [5,7,8,24], Ann Sarah Walker[1,5,6,24] & the COVID-19 Infection Survey team*

During the COVID-19 pandemic, numerous SARS-CoV-2 infections remained undetected. We combined results from routine monthly nose and throat swabs, and self-reported positive swab tests, from a UK household survey, linked to national swab testing programme data from England and Wales, together with Nucleocapsid (N-)antibody trajectories clustered using a longitudinal variation of K-means (N = 185,646) to estimate the number of infections undetected by either approach. Using N-antibody (hypothetical) infections and swab-positivity, we estimated that 7.4% (95%CI: 7.0–7.8%) of all true infections (detected and undetected) were undetected by both approaches, 25.8% (25.5–26.1%) by swab-positivity-only and 28.6% (28.4–28.9%) by trajectory-based N-antibody-classifications-only. Congruence with swab-positivity was respectively much poorer and slightly better with N-antibody classifications based on fixed thresholds or fourfold increases. Using multivariable logistic regression N-antibody seroconversion was more likely as age increased between 30–60 years, in non-white participants, those less (recently/frequently) vaccinated, for lower cycle threshold values in the range above 30, and in symptomatic and Delta (vs. BA.1) infections. Comparing swab-positivity data sources showed that routine monthly swabs were insufficient to detect infections and incorporating national testing programme/self-reported data substantially increased detection. Overall, whilst N-antibody serosurveillance can identify infections undetected by swab-positivity, optimal use requires fourfold-increase-based or trajectory-based analysis.

To July 21, 2024, almost 776 million severe acute respiratory syndrome coronavirus 2 (SARS-CoV-2) infections have been reported worldwide[1]. Nevertheless, many infections remained undetected and therefore the actual number is thought to be substantially higher[2,3]. Serological testing can potentially provide information on undetected infections, thereby improving estimates of the number of previous infections[4–6].

Several studies have explored serological testing for SARS-CoV-2 infections by analysing either spike (S-) or nucleocapsid (N-)antibodies[6–9]. Levels of both are in most people, at least temporarily,

raised after SARS-CoV-2 infection. Because the most widely used SARS-CoV-2 vaccines target the spike protein, leading to increased S-antibody levels following vaccination, S-antibodies cannot easily be used to estimate how many people have been previously infected in populations with high vaccination rates, such as high-income countries[10,11]. N-antibodies do not directly respond to the most commonly used mRNA and adenovirus SARS-CoV-2 vaccinations[10,11]. Nevertheless, the sensitivity of N-antibodies to detect infections depends strongly on the population, the time since infection and the

A full list of affiliations appears at the end of the paper. *A list of authors and their affiliations appears at the end of the paper. ✉ e-mail: l.r.zwerwer@rug.nl

thresholds used; previous studies have reported sensitivities ranging from 40-100%[4,12–16].

Various demographical characteristics have also been shown to affect N-antibody seroconversion following infection. For instance, several studies reported higher antibody titres, and hence higher seroconversion rates, in males[17,18] and older individuals[18–20]. Other factors influencing seroconversion include presence of symptoms/disease severity[16,17,19,20], hospitalisation[17,18], ethnicity[20] and body mass index[18,19]. Moreover, while N-antibodies do not directly respond to most commonly used SARS-CoV-2 vaccinations, some studies have suggested N-antibody seroconversion might be reduced in vaccinated individuals[9,13,16]. For instance, in a randomised controlled trial examining mRNA-1273 vaccine effectiveness, only 40% (95% confidence interval (CI): 27-54%; $n = 21$) of 52 vaccination recipients showed N-antibody seroconversion after polymerase chain reaction (PCR) confirmed symptomatic infection with SARS-CoV-2 versus 93% (95%CI: 92-95%; n = 605) of 648 placebo recipients[13].

Studies aimed at generating learning from the pandemic rely on accurate estimates of infection, often inferred from PCR- and lateral flow test (LFT)-based surveillance. To assess the effectiveness of these systems, it is essential to quantify the number of infections they miss that could be identified from serology, and limitations of such serosurveillance (e.g., lower response rates among specific subgroups and with asymptomatic infections (which nevertheless can transmit onwards), impact of positivity thresholds). To our knowledge, there are no studies to date estimating the effectiveness of combining N-antibody seropositivity and PCR/LFT. Here, we therefore examine the ability of N-antibodies to identify prior (undetected by swab-positivity) SARS-CoV-2 infections in a general community-based cohort including vaccinated individuals, using clustering of longitudinal N-antibody trajectories. We define SARS-CoV-2 infections as symptomatic or asymptomatic, but with sufficient replicating virus to be detectable on PCR or lead to seroconversion through an immunological response, i.e., sufficient replicating virus that onwards transmission could be possible. Additionally, we explore reasons for lack of seroconversion after PCR-confirmed SARS-CoV-2 infection, and the impact of defining infections based on different data sources.

## Results
### Population
Between February 28, 2021 and January 30, 2022, the period when N-antibodies were assayed within the COVID-19 Infection Survey (see "Methods"), 270,686 participants provided blood samples for serological testing (Supplementary Fig. 1), median 6 per participant. The median age at first N-antibody measurement was 55 years; 54.2% participants were female, 94.0% reported white ethnicity, 26.2% a long-term health condition and 5.0% reported working in healthcare (Table 1). Respectively, 7.3%, 28.4%, 58.5% and 0.1% of participants had received 1, 2, 3 and 4 vaccinations by the end of the period in which they had N-antibodies measured (denoted their study period), with 5.7% participants remaining unvaccinated throughout (e.g., due to age, ending study participation or personal choice). We defined swab-positive infections using positive and negative PCR results from routine monthly nose and throat swabs taken for the COVID-19 Infection Survey, positive swab PCR or LFT results from the national testing programmes in England and Wales or self-reported positive swab tests (see "Methods"). We aggregated swab-positive infections into four different classes: No positive swab before or during the participant's study period (81.2%), swab-positive infection before the participant's study period only (8.5%), swab-positive infection during the participant's study period only (9.9%) and swab-positive infection before and during the participant's study period (0.5%).

### Clustering of N-antibody trajectories
To classify different types of N-antibody trajectories, we used a longitudinal variation of K-means in participants with ≥4 N-antibody measurements, ensuring that the N-antibody trajectories had sufficient information to detect SARS-CoV-2 infections. This excluded 85,040 participants (Supplementary Fig. 2), who were slightly younger (Supplementary Table 1), as well as being more likely to report fewer vaccinations, as expected since those leaving the survey before January 2022 would have both fewer vaccinations and fewer measurements. Since all N-antibody measurements were censored at the lower and upper limits of quantification (respectively, 10 ng/mL and 200 ng/mL), clustering was not performed for 85,449 participants with no evidence of a previous infection (all N-antibody levels ≤10 ng/mL) and 326 participants with evidence of a previous infection (all N-antibody measurements ≥200 ng/mL) who were simply assigned to these two respective additional clusters. We therefore applied the longitudinal variation on K-means to identify 13 clusters in the remaining 99,871 participants (Supplementary Fig. 2) using absolute values (denoted identity clustering, 'id') and using $\log_2$ values (denoted '$\log_2$').

After careful examination of these 13 clusters from the two N-antibody transformations (Supplementary Fig. 3), we grouped them

## Table 1 | Characteristics of participants with any N-antibody measurement (N = 270,686)

| N = 270,686 | | |
|---|---|---|
| Number of N-antibody measurements (median [IQR]) | | 6 [3, 8] |
| Age at last birthday (years) (median [IQR], percentiles [1, 99]) | | 55 [41, 67], [15,85] |
| Sex (%) | Female | 146,823 (54.2) |
| | Male | 123,863 (45.8) |
| Ethnicity (%) | Non-White | 16,291 (6.0) |
| | White | 254,395 (94.0) |
| Long-term health condition (%) | No | 199,636 (73.8) |
| | Yes | 71,050 (26.2) |
| Healthcare worker (%) | No | 257,115 (95.0) |
| | Yes | 13,571 (5.0) |
| Vaccination* | Not vaccinated | 15,392 (5.7) |
| | 1 vaccination | 19,795 (7.3) |
| | 2 vaccinations | 76,830 (28.4) |
| | 3 vaccinations | 158,338 (58.5) |
| | 4 vaccinations | 216 (0.1) |
| | Missing | 115 (0.0) |
| Swab-positive infections[†] (%) | No infection | 219,663 (81.2) |
| | Infection before the study period[‡] | 22,920 (8.5) |
| | Infection during the study period*** | 26,836 (9.9) |
| | Infection before and during the study period**** | 1267 (0.5) |
| Spike-antibody seropositivity** | No spike seropositivity | 262,513 (97.0) |
| | Spike seropositive before the study period | 7446 (2.8) |
| | Spike seropositive during the study period | 727 (0.3) |

*Vaccination status at the end of each participant's study period.

[†] As identified from swab test results (see "Methods").

** Before any reported vaccinations.

[‡] 314 participants had two or more swab-positive infections before their study period

*** 363 participants had two or more swab-positive infections during their study period

IQR Inter quartile range.

Note: study period defined as the time from each participant's first N-antibody measurement to their last N-antibody measurement. Participants could be in the survey before this started, see "Methods".

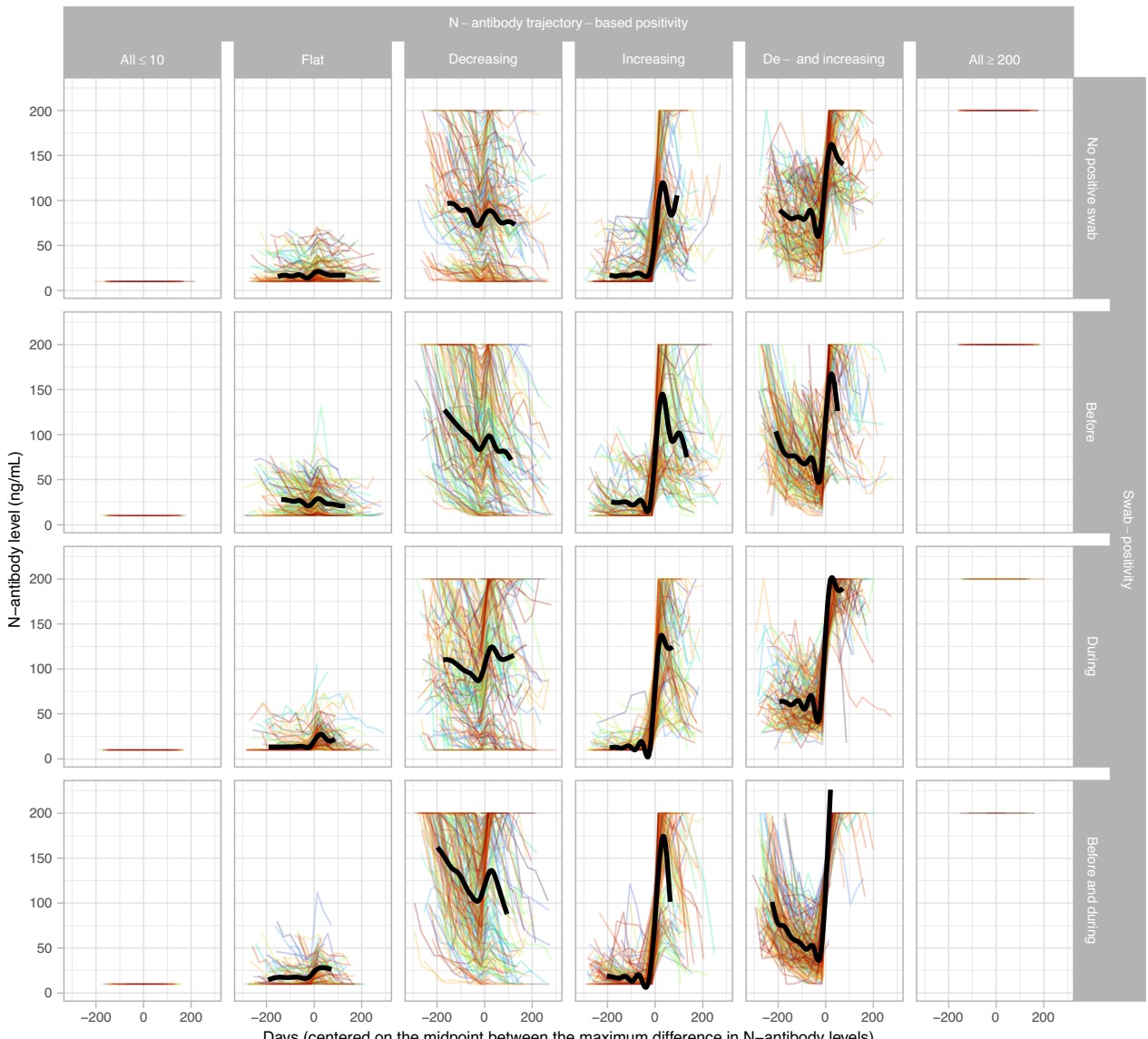

**Fig. 1 | N-antibody trajectories for the different N-antibody and swab-positive infection groups (restricted to those with ≥4 N-antibody measurements, see Supplementary Fig. 2).** For comparability, trajectories are centred on the midpoint between the maximum difference between any two consecutive measurements per participant. This approximates the hypothetical infection date for those with an N-antibody trajectory compatible with infection, but can create a small but arbitrary distortion in those without swab-positive infections and classified as flat or decreasing. Each frame contains a random sample of 200 N-antibody trajectories (see Fig. 2 for numbers and cell percentages). Black line depicts a generalised additive modelling smooth for all N-antibody measurements assayed between the 10th and 90th percentile of the centred days in each cluster.

into four types: flat, decreasing, increasing, and those that first decreased and then increased. Biologically, the different categories broadly correspond to having no evidence of an infection before or during the study period, evidence of a previous infection before the study period only, evidence of a current infection during the study period only and evidence of a previous and current infection, respectively (Supplementary Figs. 4 and 5). A final trajectory-based classification was obtained based on consensus: where the two transformations differed (N = 9644, 9.7%), often relating to smaller absolute increases which were magnified on the relative (log) scale, participants were classified using visualisation of the trajectories (Supplementary Figs. 6 and 7). Interestingly, the N-antibody trajectories for 54 participants in cluster 13 using identity clustering and cluster 10 using log2 transformed clustering implied two different infections during the participant's study period (Supplementary Fig. 7). 20 (37.0%) of these participants had two or more swab-positive

infections during their study period (compared to 350 (0.2%) among all participants with ≥4 N-antibody measurements).

Figure 1 shows the N-antibody trajectories for the final different trajectory-based classifications and swab-positive infection groups. More specifically, it shows that flat N-antibody trajectory-based classifications with no positive swab before or during the study period had relatively little variation. Flat N-antibody trajectory-based classifications with a swab-positive infection before the participant's study period only had a marginal decrease in N-antibody levels overall. Moreover, N-antibody trajectories classified as flat with a swab-positive infection during or before and during the participant's study period had a marginal increase in N-antibody levels overall. In contrast, N-antibody trajectories classified as decreasing with no positive swab before or during the study period or a swab-positive infection before their study period showed a marked decrease in N-antibody levels. Decreasing N-antibody trajectory-based classifications with a swab-

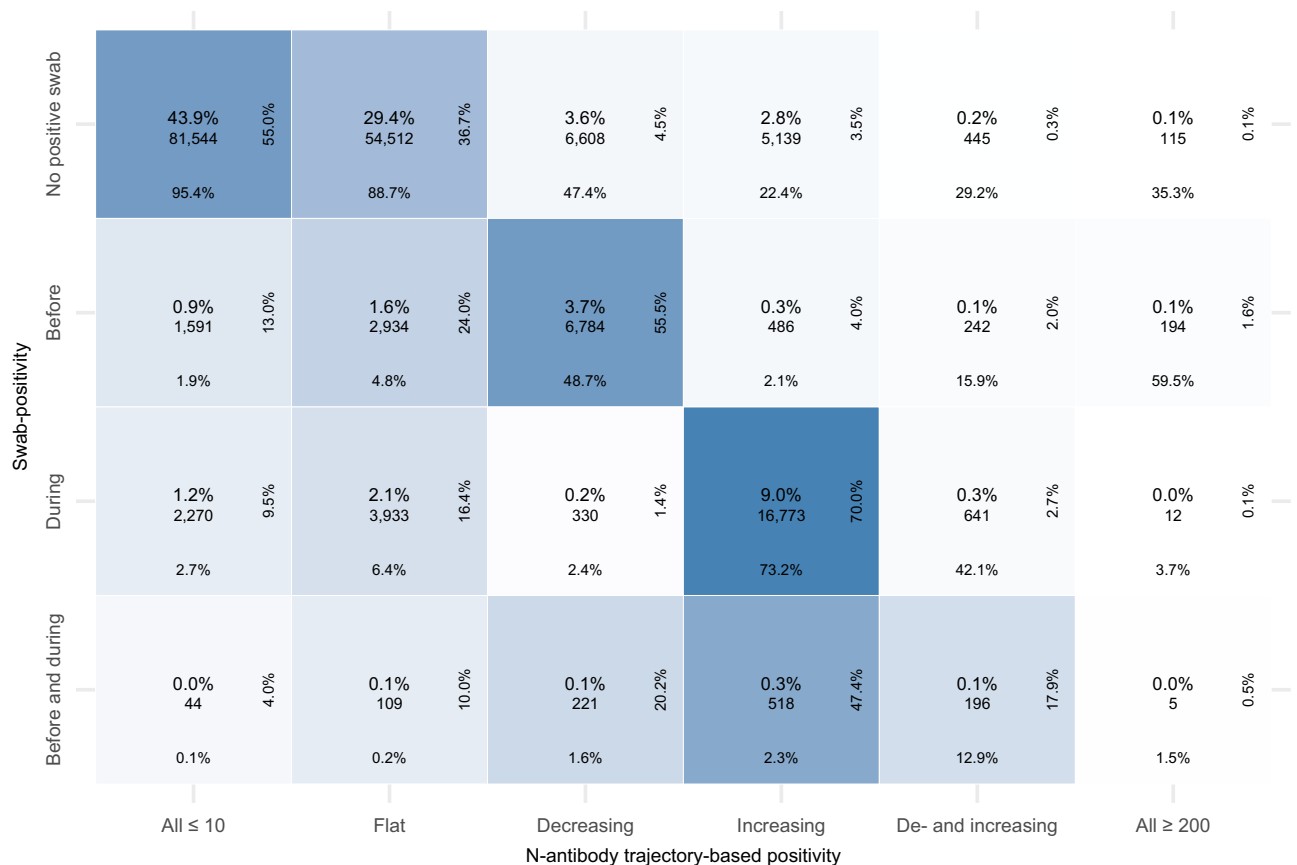

**Fig. 2 | Number of participants classified in each N-antibody trajectory-based and swab-positive infection group.** Colour intensity of the tiles indicates the row-percentage. Of all 13,324 participants with a swab-positive infection (i) before or (ii) before and during their study period, only 108 (0.8%) participants had two distinct swab-positive infections before their study period and of all 25,052 participants with a swab-positive infection (i) during or (ii) before and during their study period, 350 (1.4%) participants had two or more swab-positive infections during their study period. Note: showing raw counts, total percentages, row and column percentages.

positive infection during or before and during the participant's study period had decreasing then increasing N-antibody levels. Finally, regardless of swab-positivity group, all trajectories classified as increasing or decreasing and increasing had considerable increases in N-antibody levels.

Figure 2 shows the number of participants in the different final trajectory-based N-antibody classifications and swab-positive infection groups. Overall agreement between the N-antibody trajectory-based classification and swab-positive infections was 86.2% (95%CI: 86.0–86.3%) in all participants with ≥4 N-antibody measurements. For 73.3% (73.1–73.5%) of all participants no infection was detected by swab-positivity and N-antibody trajectory-based analysis (N-antibody trajectory-based classifications: all ≤10 or flat). 4.1% (4.0–4.2%) of participants were swab-positive before or before and during their study period and had decreasing, de- and increasing N-antibody trajectories or all N-antibody measurements equal or above 200, consistent with prior infection. For 18,128 (9.8%; 9.6–9.9%) participants, a swab-positive infection occurring during their study period was also detected by N-antibody trajectory-based analysis. For these participants, we estimated N-antibody (hypothetical) infection dates as 14 days before the midpoint between the two measurements with the maximum increase in N-antibody levels. Overall, most (61.5%) N-antibody (hypothetical) infection dates were within 15 days of the closest swab-positive date (Supplementary Tables 2 and 3, Supplementary Fig. 8), being ≥60 days in only 505 (2.8%) participants. However, 28.6% (28.1–29.2%) of the 25,404 swab-positive infections during the study period did not show any evidence of an infection from

their N-antibody trajectories. Moreover, 25.8% (25.3–26.4%) of the 24,440 participants with increasing/de- and increasing N-antibody trajectories had no evidence of a swab-positive infection.

## Estimated number of true infections

Using both N-antibody trajectory-based classifications and swab-positive infections (including multiple swab-positive infections per participant), we identified 31,716 infections during the study period in 31,364/185,646 (16.9%) participants with ≥4 N-antibody measurements (Table 2). 24,440 (77.1%) of these detected infections were identified using N-antibody trajectory-based analysis, 25,404 (80.1%) were detected with swab-positivity and 18,128 (57.2%) were detected with both swab-positivity and N-antibody trajectory-based analysis. Assuming that both types reflected true infections and there were no false-positives, using a method dependent capture-recapture model we estimated the true total number of infections during the study period among all participants with ≥4 N-antibody measurements (i.e., those detected and undetected with either N-antibody trajectory-based classifications or swab-positivity) as 34,249. Of these infections 7.4% remained undetected with either method, 25.8% by swab-positivity and 28.6% by N-antibody trajectory-based classification. Hence, assuming missed infections were singletons and that they only occurred for participants without an infection detected by either N-antibody trajectory-based analysis or swab-positives, 18.3% of participants with ≥4 N-antibody measurements would have been infected during the study period.

When stratifying by vaccination status, using a method dependent capture-recapture model, we estimated that 4.8–10.9% of the true

**Table 2 | Estimated number of true infections using different definitions of N-antibody seropositivity**

| | N-antibody seropositivity defined by | | | | |
|---|---|---|---|---|---|
| | Trajectory | Trajectory (sensitivity reclassifying ≥60 days between infection dates) | Fixed threshold | Fourfold-rise ( >200 ng/mL=200) | Fourfold-rise ( >200 ng/mL =infection) |
| Total observed infections (N-antibody or swab-positivity) | 31,716 | 31,716 | 39,511 | 30,047 | 31,577 |
| N (%) (95% CI) identified by N-antibody | 24,440 77.1% (76.6–77.5%) | 24,139 76.1% (75.6–76.6%) | 32,702 82.8% (82.4–83.1%) | 20,999 69.9% (69.4–70.4%) | 23,161 73.3% (72.9–73.8%) |
| N (%) (95% CI) identified by swab-positivity | 25,404 80.1% (79.7–80.5%) | 25,200 79.5% (79.0–79.9%) | 25,404 64.3% (63.8–64.8%) | 25,404 84.5% (84.1–85.0%) | 25,404 80.5% (80.0–80.9%) |
| N (%) (95% CI) identified by both | 18,128 57.2% (56.6–57.7%) | 17,623 55.6% (55.0–56.1%) | 18,595 47.1% (46.6–47.6%) | 16,356 54.4% (53.9–55.0%) | 16,988 53.8% (53.2–54.3%) |
| Estimated true infections* (95% CI) | 34,249 (34,115–34,383) | 34,517 (34,374–34,663) | 44,676 (44,460–44,874) | 32,615 (32,477-32,753) | 34,634 (34,482 – 34,784) |
| % undetected by both methods (95% CI) | 7.4% (7.0–7.8%) | 8.1% (7.7–8.5%) | 11.6% (11.1–12.0%) | 7.9% (7.5–8.3%) | 8.8% (8.4–9.2%) |
| % undetected by N-antibody (95% CI) | 28.6% (28.4–28.9%) | 30.1% (29.8–30.4%) | 26.8% (26.4–27.1%) | 35.6% (35.3–35.9%) | 33.1% (32.8 – 33.4%) |
| % undetected by swab-positivity (95% CI) | 25.8% (25.5–26.1%) | 27.0% (26.7–27.3%) | 43.1% (42.9–43.4%) | 22.1% (21.8–22.4%) | 26.7% (26.3 – 27.0%) |

*using method dependent capture-recapture models.

infections were undetected with either N-antibody trajectory-based classifications or swab-positivity, with respectively 6.6% (95% CI 5.1–8.2%) and 10.9% (9.9–11.9%) of all infections remaining unidentified in unvaccinated participants and participants with 3 or 4 vaccinations (Supplementary Table 4). Moreover, respectively 59.7% (50.6–68.4%), 5.8% (5.4–6.2%) and 7.4% (6.8–8.1%) of all true infections were undetected by either method during the Alpha, Delta and BA.1 epoch (Supplementary Table 4).

As a sensitivity analysis, we reclassified the 505 participants with ≥60 days between the N-antibody (hypothetical) infection date and closest swab-positive infection and estimated the percentage of undetected infections with either N-antibody-based classifications or swab-positivity. Where the swab-positive infection date was ≥60 days before the N-antibody (hypothetical) infection date, we classified the infection as detected by swab-positivity only, and as N-antibody only when the swab-positive infection date was ≥60 days after the N-antibody (hypothetical) infection date. Under these assumptions, of all detected infections, 24,139 (76.1%) were detected using N-antibody trajectory-based analysis, 25,200 (79.5%) using swab-positivity and 17,623 (55.6%) by both methods (Table 2). Under the assumption that neither method identifies any false positives, using a method dependent capture-recapture model we estimated a total of 34,517 true infections during the study period, 8.1% of which would have been undetected by both swab-positivity and trajectory-based N-antibody positivity.

Subsequently, we performed a sensitivity analysis using the manufacturer's proposed N-antibody seropositivity threshold of 30 ng/mL[21] to define N-antibody (hypothetical) infections rather than trajectory-based analysis. Using both N-antibody threshold-based classifications and swab-positive infections there were 39,511 detected infections (Table 2), of which 32,702 (82.8%) were identified using this fixed 30 ng/mL threshold. However, a much smaller percentage (25,404, 64.3%) were swab-positive; 18,595 (47.1%) infections were detected by both methods. Hence, under the assumption of no false positives as above, using a method dependent capture-recapture model, we estimated a total of 44,676 true infections during the study period, of which 11.6% would have been missed by both swab-positivity and infections defined by the fixed N-antibody threshold (compared to 7.4% of 34,249 true infections using swab-positivity and N-antibody trajectory-based classification).

When defining N-antibody (hypothetical) infections based on an arbitrary fourfold rise in consecutive antibody levels[22] (treating values above or below the limits of detection as equal to these limits, see "Methods"), there were 30,047 (hypothetical) N-antibody or swab-positive infections (Table 2). 20,999 (69.9%) were identified using the fourfold criterion and 25,404 (84.5%) using swab-positivity; 16,356 (54.4%) were detected by both. Hence, under the assumption of no false positives as above, using a method dependent capture-recapture model, we estimated that 7.9% of a total of 32,615 true infections would have remained undetected by both swab-positivity and the fourfold N-antibody seropositivity criterion.

Finally, in a sensitivity analysis, we treated values rising to ≥200 ng/mL as automatically meeting the fourfold criteria, since in the main analysis treating these as = 200 ng/mL antibody levels above 50 ng/mL could never rise to levels classified as infected. In this sensitivity analysis we identified 31,577 (hypothetical) infections (Table 2) of which 23,161 (73.3%) were identified by N-antibody and 25,404 (80.5%) by swab-positivity; 16,988 (53.8%) were identified by both. Hence, assuming no false positives (as above), we estimated a total of 34,634 true infections of which 8.8% would remained undetected by both methods.

## Associations with lack of N-antibody response
Subsequently we compared participant characteristics between swab-positive infections with (i) increasing or (ii) de- and increasing N-antibody trajectories (i.e., responders) and flat or decreasing N-antibody trajectories (i.e., non-responders) (Supplementary Fig. 9, Supplementary Table 5). In a multivariable model, we found significantly lower odds of non-response (i.e., higher odds of seroconversion) as age increased between 30 and 60 years and in non-white participants (Supplementary Table 6). We also found that vaccination influenced N-antibody non-response, with significantly lower odds of non-response in unvaccinated participants, and those that were less recently vaccinated or had fewer vaccinations. Furthermore, higher cycle threshold (Ct) values in the range above 30 were associated with significantly greater odds of non-response. Additionally, participants with symptoms were significantly less likely to be non-responders. Finally, compared to infections during the Delta epoch, infections during the BA.1 epoch were significantly more likely to be N-antibody non-responders.

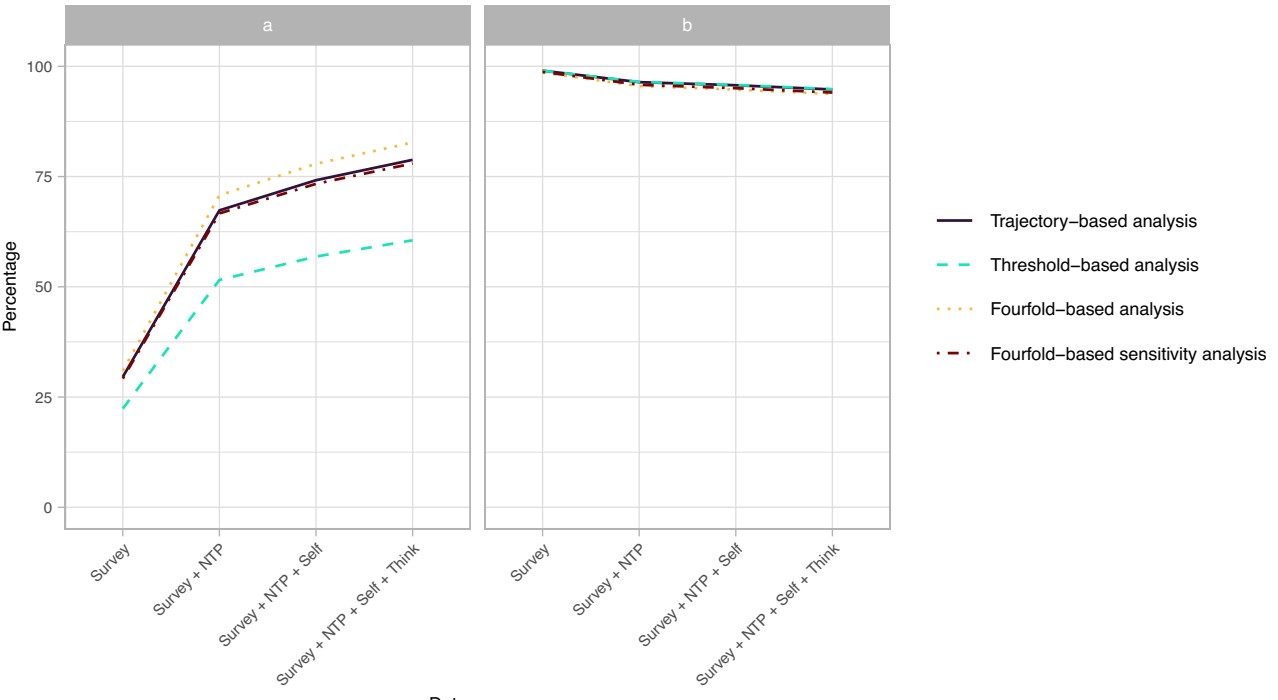

**Fig. 3 | Comparison of the N-antibody trajectory-based classification, fixed 30 ng/mL classification, fourfold-based classification and fourfold-based sensitivity analysis across different data sources used to define swab-positive infections. a** Percentage of participants with N-antibody (hypothetical) infections that were identified using swab-positivity (**b**) Percentage of participants without N-antibody (hypothetical) infections with no positive swab. In Fig. 3b the four lines (nearly) overlap. Survey: only using positive and negative swab PCR test results from the COVID-19 Infection Survey to define swab-positive infections; NTP: using positive PCR and LFT swab test results from national testing programmes in England and Wales; Self: using self-reported positive swab test results; Think: using self-report on thinking one had had COVID-19.

## Using different data sources to define infections and trajectory-based vs. fixed threshold-based vs. fourfold-based positivity

Next, we compared positivity based on N-antibody trajectories, the fixed 30 ng/mL threshold, and both the fourfold increases in N-antibody levels using different data sources to define swab-positive infections. Across the different data sources, the percentage of participants with N-antibody (hypothetical) infections that were identified using swab-positivity ranged between 29.6–78.8% for the trajectory-based classification, between 22.4-60.5% for the fixed 30 ng/mL threshold, between 30.9–82.7% for the main fourfold-based classification and between 29.2–78.0% for the fourfold-based sensitivity analysis classification (Fig. 3a). Using only results from the routinely scheduled swabs (i.e., from the COVID-19 Infection Survey alone), identification of infection was poor with between 69.1-77.6% of N-antibody responders remaining unidentified. The percentage identified through swab-positivity showed the highest increase moving from defining swab-positive infections using the survey only to the survey plus the national testing programme from England and Wales (covering 89% of participants).

Notably, there was a considerable difference in the percentage of swab-positive infections identified among N-antibody (hypothetical) infections comparing the trajectory-based classification and the threshold-based classification, with the threshold-based classification identifying consistently more participants as having N-antibody (hypothetical) infections during their study period and a lower proportion of these infections being swab-positive (Table 2, also Supplementary Table 7). The differences between both fourfold-based classifications and the trajectory-based classifications were much smaller; the main fourfold-based classification identified slightly fewer N-antibody (hypothetical) infections overall (Table 2) but a slightly higher proportion of these were swab-positive across all data sources (Fig. 3a).

The percentage of all N-antibody negative participants with no positive swab decreased from 99.0% to 94.7% with increasing richness of data source for the trajectory-based N-antibody classification, from 99.0% to 94.8 % for the threshold-based classification, from 98.6% to 93.7% for the fourfold-based N-antibody classification and from 98.7% to 94.1% for the fourfold-based sensitivity analysis (Fig. 3b). Interestingly, including participants thinking they had had COVID-19 as a positivity criterion (without any swab-positivity) made only marginal differences in the percentage of participants without (swab)-positive infections among N-antibody negatives.

Supplementary Fig. 10 visualises the trajectories of N-antibody negative and N-antibody positive participants using the main trajectory-based and alternative threshold-based classifications, stratified by swab-positivity (all during the participant's study period). It shows that most participants with N-antibody negative trajectory-based and N-antibody positive threshold-based classification had no positive swab (87.4%) and the N-antibody trajectories mostly had a marginal increase, to just above the 30 ng/ml threshold. Trajectories from participants with N-antibody positive trajectory-based and negative threshold-based classifications had large increases in N-antibody levels but 59.5% still had no positive swabs during their study period.

In contrast, comparing main trajectory-based and alternative fourfold-rise-based classifications, smaller numbers of participants had positive fourfold-based and negative trajectory-based N-antibody classification (main N = 281, Supplementary Fig. 11; sensitivity analysis N = 1810, Supplementary Fig. 12). However, most of these (hypothetical) infections were swab-negative (68.0% and 84.3%, respectively),

with generally decreasing N-antibody trends before the (hypothetical) infection, consistently with either a single falsely low measurement or prior infection. In contrast, the larger number of negative fourfold-based and positive trajectory-based N-antibody classification showed an evident increase in N-antibody levels in both main and sensitivity analysis (Supplementary Figs. 11 and 12). These N-antibody increases were either slightly blunted or started from higher initial levels and so did not reach a fourfold increase. Only 50.0% and 46.1% of these participants respectively were swab-positive during their study period.

## Discussion

During the recent COVID-19 pandemic, hundreds of millions of individuals tested positive for a SARS-CoV-2 infection. However, due to a considerable number of asymptomatic individuals, the true number of infections remains unknown[2]. In this study we used data from a large broadly representative UK household survey and examined the efficacy of detecting prior (undetected) SARS-CoV-2 infections in the general population by clustering of N-antibody trajectories. We found that under the assumption that swab-positives and N-antibody positives both reflect true infections, 7.4% of all true SARS-CoV-2 infections (i.e., those detected and undetected by swab-positivity and N-antibody trajectory-based classifications) would have remained unidentified from both swab results and N-antibody trajectories (compared to 25.8% by swab-only and 28.6% by trajectory-based N-antibody classifications only).

As far as we are aware, no other study has examined the efficacy of the combination of swab-positivity and N-antibody serological testing to identify SARS-CoV-2 infections and used this combination to estimate the number of infections remaining undetected by either method, particularly not on this scale. However, several studies have examined the ascertainment rate for swab-positivity alone in the UK. For instance, Colman et al. (2023) estimated that, after SARS-Cov-2 testing became widely available in the UK, 60-70% of all infections remained undetected by national healthcare and community testing programmes by calibrating reported cases to the swab-positivity rate from the COVID-19 infection survey, while accounting for the incubation period distribution, and the time-dependent test sensitivity of PCR and lateral flow tests[23]. Nightingale et al. (2022) estimated the under-ascertainment rate at 75.0% using swab-positivity from the COVID-19 infection survey as a ground truth to estimate the performance of the national testing programmes[24]. Here, we took a different approach and estimated the 'true' number of infections by applying a log-linear capture-recapture model to infections detected by positive swab tests – from the COVID-19 infection survey or national testing programmes, or self-reported positive swab test – or infections detected through the clustering analysis performed on N-antibody results. Focusing purely on the swabs included in this comparison, 25.8% of all infections were missed. Further, we found that estimates of undetected infections varied substantially across virus variants, with considerably more undetected infections during the Alpha epoch. During the Alpha epoch only one third of all detected infections was ascertained by swab-positivity (vs. 76.8% and 88.4% during the Delta and BA.1 epoch, see Supplementary Table 4). These differences may have been caused by the limited availability of testing during the Alpha epoch (i.e., in the UK LFT were made widely available in April 2021, at the end of the Alpha epoch[25]). Earlier research also showed that estimates of undetected infections vary by age group, variant, and region and that these variations may be related to differences in symptoms/disease severity, public sentiment and availability of testing[23].

Notably, even using these increasingly rich data sources on swab-positivity, still 25.8% of all true infections remained undetected by swab-positivity during the study period. This may be for a variety of reasons. Firstly, the timing of the swab is of critical importance and can result in false-negative results[26,27]. Monthly swabs in the survey were a trade-off between expected duration of PCR positivity (mean 21 days

from infection in human challenge studies[28]) and costs, aiming to identify >80% of infections. Secondly, viral load can be unequally distributed throughout the body[3], with higher viral RNA concentrations in stool and sputum[29], and related to severity of symptoms[30]; low viral load could cause false-negative swab results[27]. Other reasons for false-negative swabs include viral genetic variation and challenges with self-sampling[27].

Consistent with a much smaller study among hospitalised individuals, we found that a little over a quarter of all swab-positive infections did not seroconvert in terms of N-antibodies[4]. However, several other studies also report lower percentages of non-responders among swab-positives[12,14,16,18,19,31,32]. Nevertheless, these studies did not use N-antibody trajectories to define seroconversion, which made substantial differences in identification, and predominately focused on specific subgroups, such as healthcare workers and/or had small sample size. Consistent with most other studies, we found no association between seroconversion rates and gender[9,16,19,20]. We found N-antibody seroconversion rates increased as age increased between 30 and 60 years, consistent with higher antibody titres (and thus higher seroconversion rates) in older individuals in some studies[18–20], although one study found higher seroconversion rates among younger age groups compared to individuals ≥65 years (adjusted for vaccination)[16]. Again consistent with the literature, we found seroconversion was more likely among individuals who reported non-white ethnicity[20], were less (recently or frequently) vaccinated[9,13,16], infections with lower Ct values in the range above 30[20] (a proxy for viral load[33]) and symptomatic infections[19,20]. However, in contrast to one previous study, we also found that participants with an infection during the BA.1 epoch were significantly less likely to seroconvert compared to participants with an infection during the Delta epoch[16]. This could potentially relate to the low number of asymptomatic infections in this previous study[16], since the proportion of asymptomatic infections is significantly higher for Omicron compared to Delta infections[34] and we, and others, have shown that individuals with asymptomatic infections are less likely to seroconvert[19,20].

Where participants were identified as having been infected using both approaches, estimated N-antibody (hypothetical) infection dates were mostly within 15 days of the closest swab-positive date. Nonetheless, the percentage of N-antibody (hypothetical) infections identified using swab-positivity was highly dependent on the data source. We incrementally tested adding the different data sources into swab-positivity definitions, reflecting their likely level of ascertainment. Using the survey swab-positivity alone, only approximately a quarter of all N-antibody trajectory-based infections were identified. The use of data from national testing programmes vastly increased infection identification rates, although on their own, they provide a poor level of ascertainment (as above[23,24]) and incorporation of unbiased swab positivity testing data from the COVID-19 infection survey has been demonstrated to be essential to reconstruct the epidemic[35]. Using 'thinking one had COVID-19' as a positivity criterion only modestly increased the number of N-antibody infections identified, whilst having a marginal impact on the percentage of false-negatives, which is remarkable considering that an earlier study showed that in the UK only 51.5% of all individuals recognises common COVID-19 symptoms[36]. Compared to threshold-based N-antibody positivity classifications (based on the manufacturer's threshold), trajectory-based classification was consistently more aligned with swab-positivity. The threshold-based classification identified considerably more (hypothetical) infections whose trajectories were relatively flat but elevated and never tested positive by swab. These relatively flat but elevated antibody trajectories could potentially reflect cross-reactivity[10,37]. Differences between trajectory-based and fourfold-rise-based N-antibody classifications were much smaller (Fig. 3, Table 2). While only 50.0% of the trajectory-based (hypothetical) N-antibody infections and fourfold-rise-based seronegatives were swab-positive,

the N-antibody measurements showed a clear increase in N-antibody levels (Supplementary Fig. 11), with N-antibodies starting from higher levels or responses being slightly blunted. Hence, while the fourfold-based N-antibody classifications were slightly better aligned with swab-positivity, trajectory-based classifications could offer advantages when blunted N-antibody responses might be expected (e.g., in vaccinated populations) or when N-antibody levels are higher (e.g., in previously infected populations), and are less susceptible to single erroneous measurements. Furthermore, whilst based on a previous study[22], the fourfold rise was an arbitrary threshold.

The main study strength is our use of a longitudinal variation of K-means to identify infections from N-antibody trajectories. The challenge of using an arbitrary fixed threshold or a x-fold increase is that the boost in antibodies following infection is not consistent between individuals; for different reasons, some individuals will have a larger or smaller increase, from a higher or lower initial starting value. Both types of heterogeneity cause problems for simple definitions based on fixed or relative thresholds, problems which are magnified by censoring by limits of detection (here 10 and 200 ng/mL). N-antibodies were generally assayed monthly and comparing how antibody levels changed over time allowed us to still classify participants with "blunted" responses as having been infected and also taking into account declines over time which could affect sensitivity of fixed or relative thresholds. However, our study has several limitations. Firstly, N-antibody measurements were obtained using one assay only, which was ultimately not commercialised. Secondly, we only applied one clustering method and due to computational limitations were not able to optimise the number of clusters. However, in contrast to most other studies that aim to cluster a high-dimensional space, we clustered time-series, which allowed for visualisation and thorough inspection of clusters without projection methods that depend on hyperparameters and interpretation such as Uniform Manifold Approximation and Projection[38]. Moreover, swab-positivity allowed careful triangulation of each cluster, overall leading to a biologically plausible classification for most participants (Figs. 1, 2). While overall the clustering led to sensible infection groups, several antibody trajectories classified as flat by the trajectory-based N-antibody classification exhibited slight variations in antibody levels, plausibly due to the centreing of the antibody trajectories creating small but arbitrary distortions. Other explanations include variability in the assay, which is more pronounced in unspiked serum/plasma[21], prior infections before the participant's study period and mild infection episodes or infection episodes in participants with compromised immune systems. Nevertheless, for those participants without swab-positives it remains uncertain if these variations were caused by such external influences or actual infections. Next, by necessity all measurements below or above the lower and upper limits of quantification were censored, potentially leading to incorrect N-antibody trajectory-based classifications. For instance, fully censored participants could have had a considerable increase or decrease in N-antibody measurements, which was no longer visible due to the censoring. Furthermore, the number of participants with a SARS-CoV-2 infection before their study period is most likely an underestimation of the true number of infections, given lack of widespread testing in the first wave in March–May 2020, and recruitment of most survey participants from July–October 2020. Also, detection of previous infections using N-antibodies depends on the durability of seropositivity, with S-antibody response (before vaccination) in general more persistent than N-antibody response[31]. Estimating the percentage of infections that remained undetected by swab-positivity and N-antibodies depended on PCR tests not being subject to false-positives and N-antibody trajectories not being subject to cross-reactivity. Previous analyses using the COVID-19 Infection Survey have shown that specificity of the PCR testing protocol was really high, alleviating concerns about potential false-positives resulting from PCR testing[39]. Specificity has also been suggested to be very high for the N-antibody tests[40]. Moreover, we used a method dependent capture-recapture model, in which the probabilities of detecting true infections varied by method, but not per individual and infection episode, which over simplifies reality as seen in the subgroup analysis (Supplementary Table 4). Data from national testing programmes in Northern Ireland and Scotland were not available (11% of survey population); to mitigate this we also included self-reported positive swab results which had very high agreement with national testing data in England and Wales (>95%). Finally, we had no information on symptom severity, which could also be related to N-antibody seropositivity[19].

The longitudinal variation of K-means applied here could also be evaluated for other antibodies/infections to assess whether surveillance of immunity and infection incidence could be improved, but its application to smaller studies has several limitations. Firstly, each participant should have sufficient measurements to detect infections (e.g., here ≥4 measurements). Moreover, K-means can be sensitive to outliers, which can have large influence in smaller datasets[41], leading to less robust clustering or possibly overfitting. For smaller datasets, other clustering algorithms less sensitive to outliers, such as K-medoids, can be more appropriate[41].

The current study showed that estimates of SARS-COV-2 infection based on PCR- and LFT surveillance or serosurveillance greatly underestimate the true burden of infection. Even combining these methods did not provide a complete overview of infection. These underestimations could bias the estimation of other parameters, such as the infection fatality ratio[42]. The results of subsequent studies simulating the pandemic or cost-effectiveness analyses may then become influenced by these biased parameters, which can lead to implementation of interventions that appear to be optimal in modelling but that lead to overall population harm when implemented in practice. Moreover, our study underlines that many SARS-CoV-2 infections remained undetected. While contributing to virus spread, these undetected infections might substantially complicate pandemic control. Our findings suggest that to optimise pandemic surveillance a combination of serosurveillance with swab-positivity should be used.

In conclusion, we used N-antibody trajectories from a large broadly representative UK household survey to examine the total number of undetected SARS-CoV-2 infections. Whilst N-antibodies serosurveillance can be used to improve estimates of the number of previous infections, for optimal use in large datasets, fourfold-based analysis or ideally trajectory-based analysis is required over threshold-based analysis.

## Methods

### Data collection

Data came from the UK's Office for National Statistics (ONS) COVID-19 Infection Survey (ISRCTN21086382, protocol on https://www.ndm.ox.ac.uk/covid-19/covid-19-infection-survey/protocol-and-information-sheets), a large longitudinal survey inviting all individuals aged 2 years or older living within randomly selected private households across the UK to participate. Following verbal consent, study workers visited each household, and recruited all consenting residents aged 2 years or older who provided written informed consent (from parents/carers for those under 16 years; those aged 10–15 years also provided written assent). Participants could also provide optional consent for subsequent weekly visits in the first month and then monthly, up to the earliest of March 2023, when they became no longer resident at the selected address or no longer wished to participate (98% consented to post-enrolment visits). Ethical approval was obtained from the South Central Berkshire B Research Ethics Committee (20/SC/0195).

Data was collected on participants socio-demographic characteristics; at each assessment, data was collected on behaviours and vaccination status, and participants provided a nose and throat swab for PCR testing (self-taken; parents/carers took swabs for those under

12 years) (details in Supplementary File 1). Initially, those aged ≥16 years from a random 10–20% households were asked for optional consent to give monthly venous blood samples for serological testing; this was expanded to a larger randomly selected subgroup of households from April 2021 using capillary blood sampling to examine vaccine responses (prioritising those with longer survey participation). Moreover, any participant ≥16 years testing PCR-positive through December 2021 was invited to provide blood samples on their subsequent monthly follow up visits.

## Serological testing and definition of infections

Levels of SARS-CoV-2 S-antibody (throughout) and N-antibody (between February 28, 2021 and January 30, 2022 to monitor initial responses to the vaccination programme) were tested on venous or capillary blood samples using an enzyme-linked immunosorbent assay (ELISA) detecting anti-trimeric spike and nucleocapsid IgG developed by the University of Oxford. Before 26 February 2021, the S-antibody assay used fluorescence detection, with a positivity threshold of 8 million units validated on banks of known SARS-CoV-2-positive and -negative samples[43]. After this, the S-antibody used a commercialised CE-marked version of the assay, the Thermo Fisher OmniPATH 384 Combi SARS-CoV-2 IgG ELISA (Thermo Fisher Scientific), with the same antigen and colorimetric detection, reporting normalised results in ng/mL of mAb45 monoclonal antibody equivalents (details in ref. [7]) and using 42 ng/mL as the threshold for an IgG-positive or -negative result (corresponding to the 8 million units with fluorescence detection). SARS-CoV-2 N-antibody levels were tested using a research-use only assay (details in ref. [21]). At the manufacturer's threshold of 30 ng/mL, the sensitivity of this assay was 94.3% and specificity 92.8%. Lower and upper limits of quantification were 10 and 200 ng/mL respectively.

The study period was defined as the period in which participants had N-antibody measurements available. All survey data after the participant's study period was excluded from this analysis. We defined 'infection episodes' using results from swab test results as in ref. [44]. In brief we used all positive and negative PCR test results from the survey, linked information about positive only PCR and LFT from the national testing programmes in England and Wales (not available for Scotland and Northern Ireland), self-reported positive swab tests from all participants (as national testing data was not available in Scotland/Northern Ireland; very high (>95%) agreement for participants in England and Wales). To reflect the fact that some individuals can test positive on PCR for extended periods of time when testing is independent of symptoms/case contacts as in the survey (in contrast to national testing programmes), whereas others have reinfections (confirmed by sequencing) after only short periods of time, we incorporated information from genetic sequencing, S-gene presence/absence, and Ct values, together with negative PCR test results from the survey only[44]. These data were processed using Stata MP 16[45].

## Classifying N-antibody trajectories

We clustered similar N-antibody trajectories in participants with ≥4 measurements together using a longitudinal variation of K-means with a dynamic time-warping distance to account for varying periods of availability of N-antibody measurements, and gaps in each participant's trajectory due to missed visits or failed assays (details in Supplementary file 1)[46–48]. This clustering method takes into account the shape and the height of the antibody trajectory. Hence, by considering both these characteristics the clustering can distinguish different clusters with similar shapes but different heights. Characteristics of those with <4 vs ≥4 N-antibody measurements were compared using standardised differences calculated as $\frac{p1-p2}{\sqrt{\frac{p1(1-p1)+p2(1-p2)}{2}}}$[49]. Participants with all N-antibody measurements either ≤10 or ≥200 were not formally clustered but assigned to two additional clusters. Due to the large

sample size, optimisation of the number of clusters was not computationally feasible. Therefore, we chose to fit the largest number of clusters which was still computationally feasible to converge within 2 days ($n = 13$, taking 40 h on 10 cores).

To reflect the fact that both absolute and relative changes in N-antibody levels might indicate infection, we clustered N-antibody trajectories firstly using absolute values (denoted identity clustering, 'id') and secondly, using $\log_2$ values (denoted '$\log_2$'). Five different initialisations were used for each, with a maximum of 50 iterations, returning the clustering solution with the lowest sum of squared dynamic time-warping distances between each trajectory and the corresponding cluster centroid (i.e., minimal inertia)[48]. The clustering analysis was performed in Python version 3.10.12[50] (jupyter-notebook version 7.1.2) using the packages pandas (version 1.4.2), numpy (version 1.23.5), tslearn (version 0.5.2)[48] and dill (version 0.3.4). All subsequent data processing steps and visualisations were performed in R version 4.3 using tidyverse (version 2.0.0), dplyr (version 1.1.4) and ggplot2 (version 3.5.0)[51,52].

The N-antibody trajectories in each cluster were visualised together with a generalised additive model smooth (function 'geom_smooth(method ='gam')' from ggplot2[51]). The smoothing function was applied to all days (centred on the midpoint of the maximum increase in N-antibody levels) falling between the 10th and 90th percentile. We arbitrarily classified the N-antibody trajectories based on expected trajectories following infection (Supplementary Fig. 3) and then took the consensus of the id and $\log_2$ N-antibody classifications, with manual reconciliation where these disagreed (see Supplementary Fig. 2, 6 and 7 and Results), and compared the combined final classification with swab-positive infections as defined above (Figs. 1 and 2).

## Estimating infection dates

For participants with an N-antibody trajectory compatible with infection, we estimated the (hypothetical) infection date (the first date a participant would have tested positive on a nose and throat swab) assuming that the infection occurred 14 days before the midpoint between the two measurements with the maximum increase in N-antibody levels, given it takes on average ten days for N-antibodies to rise after developing symptoms[8], the incubation period is ~6.5 days[53] and on average it takes 2.5 days from infection to swab-positivity[54]. We then compared this (hypothetical) infection date estimated from N-antibody measurements with actual swab-positive infection dates (as defined above) for all participants with infections identified using both methods. Where participants had multiple swab-positive infections, we compared the closest swab-positive infection date to the N-antibody (hypothetical) infection date.

## Estimating the total number of infections

To estimate the number of true infections in those participants with ≥4 N-antibody measurements, we used a capture-recapture model[55]. This technique fits a loglinear model to the number of infections identified by swabs, N-antibody trajectory-based classifications and their intersection to estimate the number of infections missed by either methods. To reflect the fact that the number of true infections was equal for both methods, we used a closed population model. We accounted for heterogeneity in the infection detection probabilities of swabs and N-antibody trajectories by fitting a method dependent capture-recapture model, which allows the probabilities of detection to vary for swabs and N-antibody trajectories. To prevent overfitting, we chose not to model heterogeneity between infection episodes, meaning that all infection episodes had the same probability of being detected within each method. For participants with multiple swab-positive infections, we considered the closest swab-positive infection to the N-antibody (hypothetical) infection date detected by both methods and all other swab-positive infections detected by swab-positivity only. Moreover, we assumed that both swab-positives and N-antibody

(hypothetical) infections reflected true infections (i.e., no false-positives).

We performed a subgroup analysis in which we calculated the number of true infections for different vaccination statuses and epochs. Both were determined at time of the infection, which was at the swab-positive date when available and otherwise at the N-antibody (hypothetical) infection date. Dependent on the time of infection the SARS-CoV-2 epoch was defined as Alpha when it was between December 7, 2020–May 16, 2021, Delta between May 17, 2021–December 12, 2021, and BA.1 between December 13, 2021–February 20, 2022, which was the first Monday where S-positivity for the corresponding variant was above 50% in the full survey population.

Next, we performed four sensitivity analyses. Firstly, we reclassified all participants with ≥60 days between the estimated infection dates from the two methods. Where the swab-positive date was ≥60 days before the N-antibody (hypothetical) infection date, we classified the infection as swab-positivity only, and as N-antibody only when the swab-positive infection date was ≥60 days after the N-antibody (hypothetical) infection date. Secondly, we classified N-antibody trajectories using the manufacturer's proposed N-antibody seropositivity threshold of 30 ng/mL[21]. We considered a participant infected if the manufacturer's threshold of 30 ng/mL was crossed (upwards) during their study period. In all other scenarios, we considered a participant not infected during their study period, which could either mean they had an infection prior to their study period (i.e., first measurement ≥30 with no subsequent rise from < 30 to ≥30 ng/mL) or their N-antibodies measurements showed no evidence of an infection at all (all measurements < 30 ng/mL). Consistent with the trajectory-based analysis, and given the small number of participants with multiple swab-positives (350, 0.2%), we did not try to identify multiple N-antibody (hypothetical) infections during the study period. Thirdly, at the suggestion of a reviewer, in a sensitivity analysis we considered participants infected during the study period if their N-antibody levels rose fourfold over consecutive measurements[22], treating values below or after the limits of detection (10 and 200 ng/mL respectively) as equal to these limits. Lastly, we performed a sensitivity analysis on these fourfold-based classifications, treating antibody levels rising from below to ≥200 ng/mL as infected (since otherwise no value starting above 50 ng/mL could ever be considered an infection, regardless of the rise). Consistent with the trajectory-based analysis, we did not try to identify multiple N-antibody (hypothetical) infections in both fourfold-based analyses. All capture-recapture models were fitted in R version 4.3 using the package Rcapture (version 1.4.4)[55].

### Associations with participant characteristics

We investigated lack of N-antibody seroconversion amongst participants with swab-positive infections and ≥4 N-antibody measurements (in whom seroconversion could be assessed as above) using logistic regression including all demographics and information related to the infection as covariates, that is age, sex, ethnicity, healthcare worker, long-term health condition, vaccination status at time of the swab-positive infection, Ct values (a proxy for viral load[33]), symptoms and the SARS-CoV-2 epoch (complete case analysis; details in Supplementary Fig. 9; results for other covariates similar excluding Ct values (most missing data) from the model). Participants with a N-antibody (hypothetical) infection and ≥60 days between the two infection dates were excluded from this analysis, as were a small number of participants with an earlier infection identified only by S-antibody seropositivity, as this could possibly be a marker of (previous) unregistered vaccination. We additionally excluded a very small number of infections before May 17, 2021 (emergence of Delta) (N = 104/17,419 (0.6%)).

For participants with increasing/de- and increasing N-antibody trajectories who had multiple swab-positive infections, we considered the closest swab-positive infection to the N-antibody (hypothetical) infection date a responder and all other swab-positive infections non-responders. Vaccination was considered at the swab-positive infection date. Since there was limited variability in the time since vaccination for participants with 1, 3 and 4 vaccinations at the swab-positive infection (i.e., <250 participants were vaccinated >3 months ago), we aggregated time since vaccination and number of vaccinations into 7 different vaccination categories: not vaccinated, 1 vaccination, 2 vaccinations ≤3 months ago, 2 vaccinations 3–6 months ago, 2 vaccinations >6 months ago and 3 or 4 vaccinations, ignoring vaccinations ≤14 days before the swab-positive infection date. Dependent on the swab-positive infection date the SARS-CoV-2 epoch was again defined as Alpha when it was between December 7, 2020–May 16, 2021, Delta between May 17, 2021–December 12, 2021 and BA.1 between December 13, 2021–February 20, 2022. We initially fitted models with smooths for continuous covariates (age, Ct values) in R version 4.3 using nnet (version 7.3.19) and splines (version 4.3.2); for interpretability, final models used piecewise linear effects with knots chosen based on visualisations of these smooths. The final models were fitted using the glm() function in R. P-values for each regression coefficient were obtained using a two-sided Wald-test.

### Other definitions of infections

Finally, we compared N-antibody (hypothetical) infections with infections defined using different data sources, specifically (i) only positive (and negative) swab PCR test results from the survey, (ii) positive and negative PCR results from the survey and positive swab PCR or LFT results from national testing programmes in England or Wales, (iii) positive and negative PCR results from the survey, positive swab PCR or LFT results from national testing programmes in England or Wales, and self-reported swab-positives and (iv) positive and negative PCR results from the survey, positive swab PCR or LFT results from national testing programmes in England or Wales, self-reported swab-positives and self-reports that participants thought they had had COVID-19. For each, we estimated the percentage of swab-positive infections among those with N-antibody (hypothetical) infections (considering only the closest swab-positive infection to the N-antibody (hypothetical) infection date) and the percentage of participants without swab-positive infections among those without N-antibody (hypothetical) infections (N-antibody trajectory-based classifications: all ≤10, flat, decreasing or all ≥200). These percentages are equivalent to estimating the sensitivity and specificity of swab-positivity using N-antibody (hypothetical) infections as a reference. Sensitivity analysis used classifications based on the manufacturer's threshold, and both fourfold-based N-antibody classifications.

### Reporting summary

Further information on research design is available in the Nature Portfolio Reporting Summary linked to this article.

## Data availability

De-identified study data are available for access by accredited researchers in the ONS Secure Research Service (SRS) for accredited research purposes under part 5, chapter 5 of the Digital Economy Act 2017. Individuals can apply to be an accredited researcher using the short form on https://researchaccreditationservice.ons.gov.uk/ons/ONS_registration.ofml. Accreditation requires completion of a short free course on accessing the SRS. To request access to data in the SRS, researchers must submit a research project application for accreditation in the Research Accreditation Service (RAS). Research project applications are considered by the project team and the Research Accreditation Panel (RAP) established by the UK Statistics Authority at regular meetings. Project application example guidance and an exemplar of a research project application are available. A complete record of accredited researchers and their projects is published on the

UK Statistics Authority website to ensure transparency of access to research data. For further information about accreditation, contact Research.Support@ons.gov.uk or visit the SRS website.

## Code availability

A copy of the analysis code is available at: https://github.com/UMCG-Global-Health/COVID-19_N-antibodies (https://doi.org/10.5281/zenodo.13934702).

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

## Acknowledgements

We are grateful for the support of all COVID-19 Infection Survey participants. This study was funded by the UK Health Security Agency and the Department of Health and Social Care with in-kind support from the Welsh Government, the Department of Health on behalf of the Northern Ireland Government and the Scottish Government. The views expressed are those of the authors and not necessarily those of the National Health Service, NIHR, Department of Health, or UKHSA. For the purpose of Open Access, the author has applied a CC BY public copyright licence to any Author Accepted Manuscript version arising from this submission.

This work contains statistical data from ONS which is Crown Copyright. The use of the ONS statistical data in this work does not imply the endorsement of the ONS in relation to the interpretation or analysis of the statistical data. This work uses research datasets (https://doi.org/10.57906/r47r-1735) which may not exactly reproduce National Statistics aggregates.

## Author contributions

The COVID-19 Infection Survey was designed, planned, and conducted by A.S.W., K.B.P., and the COVID-19 Infection Survey Team. This specific analysis was designed by A.S.W., K.B.P., L.R.Z., and T.E.A.P., L.R.Z. conducted all statistical analyses. L.R.Z., A.S.W., and K.B.P. draughted the manuscript, and all authors contributed to the interpretation of the data and results and revised the manuscript. T.E.A.P., K.B.P., and A.S.W. contributed equally. All authors approved the final version of the manuscript.

## Competing interests

This study was funded by the UK Health Security Agency and the Department of Health and Social Care with in-kind support from the Welsh Government, the Department of Health on behalf of the Northern Ireland Government and the Scottish Government. ASW and KBP are supported by the National Institute for Health Research Health Protection Research Unit (NIHR HPRU) in Healthcare Associated Infections and Antimicrobial Resistance at the University of Oxford in partnership with the UK Health Security Agency (UK HSA) (NIHR200915). ASW is also supported by the NIHR Oxford Biomedical Research Centre. KBP is also supported by the Huo Family Foundation. There are no other conflicts of interest.

## Additional information

[1]Nuffield Department of Medicine, University of Oxford, Oxford, UK. [2]Department of Health Sciences, University of Groningen, University Medical Center Groningen, Groningen, The Netherlands. [3]Center for Information Technology, University of Groningen, Groningen, The Netherlands. [4]Department of Infectious Diseases and Microbiology, Oxford University Hospitals NHS Foundation Trust, John Radcliffe Hospital, Oxford, UK. [5]The National Institute for Health Research Health Protection Research Unit in Healthcare Associated Infections and Antimicrobial Resistance at the University of Oxford, Oxford, UK. [6]The National Institute for Health Research Oxford Biomedical Research Centre, University of Oxford, Oxford, UK. [7]Health Economics Research Centre, Nuffield Department of Population Health, University of Oxford, Oxford, UK. [8]Nuffield Department of Primary Care Health Sciences, University of Oxford, Oxford, UK. [24]These authors contributed equally: Tim E. A. Peto, Koen B. Pouwels, Ann Sarah Walker. ✉e-mail: l.r.zwerwer@rug.nl

## the COVID-19 Infection Survey team

David W. Eyre[5,6,9], Nicole Stroesser[1,4,5,6], Philippa C. Matthews[1,10,11], Jia Wei[1,6,9], Ian Diamond[12], Ruth Studley[12], Nick Taylor[12], Emma Rourke[12], Tina Thomas[12], Dawid Pienaar[12], Joy Preece[12], Sarah Crofts[12], Lina Lloyd[12], Michelle Bowen[12], Daniel Ayoubkhani[12], Russell Black[12], Antonio Felton[12], Megan Crees[12], Joel Jones[12], Esther Sutherland[12], Derrick W. Crook[1], Emma Pritchard[1], Karina-Doris Vihta[1], Alison Howarth[1], Brian D. Marsden[1], Kevin K. Chau[1], Lucas Martins Ferreira[1], Wanwisa Dejnirattisai[1], Juthathip Mongkolsapaya[1], Sarah Hoosdally[1], Richard Cornall[1], David I. Stuart[1], Gavin Screaton[1], Katrina Lythgoe[9], David Bonsall[9], Tanya Golubchik[9], Helen Fryer[9], John N. Newton[13], John I. Bell[14], Stuart Cox[14], Kevin Paddon[14], Tim James[14], Thomas House[15], Julie Robotham[16], Paul Birrell[16], Helena Jordan[17], Tim Sheppard[17], Graham Athey[17], Dan Moody[17], Leigh Curry[17], Pamela Brereton[17], Ian Jarvis[18], Anna Godsmark[18], George Morris[18], Bobby Mallick[18], Phil Eeles[18], Jodie Hay[19], Harper VanSteenhouse[19], Jessica Lee[20], Sean White[21], Tim Evans[22], Lisa Bloemberg[21], Katie Allison[22], Anouska Pandya[22], Sophie Davis[22], David I. Conway[23], Margaret MacLeod[23] & Chris Cunningham[23]

[9]Big Data Institute, University of Oxford, Oxford, UK. [10]The Francis Crick Institute, 1 Midland Road, London, UK. [11]Division of Infection and Immunity, University College London, London, UK. [12]Office for National Statistics, Newport, UK. [13]Office of the Regius Professor of Medicine, University of Oxford, Oxford, UK. [14]Oxford University Hospitals NHS Foundation Trust, Oxford, UK. [15]University of Manchester, Manchester, UK. [16]UK Health Security Agency, London, UK. [17]IQVIA, London, UK. [18]National Biocentre, Milton Keynes, UK. [19]Glasgow Lighthouse Laboratory, London, UK. [20]Department of Health and Social Care, London, UK. [21]Welsh Government, Cardiff, UK. [22]Scottish Government, Edinburgh, UK. [23]Public Health Scotland, Edinburgh, UK.

