## [Transparent Peer Review file · Nature Communications]

Identification of undetected SARS-CoV-2 infections by clustering of Nucleocapsid antibody trajectories

Corresponding Author: Ms Leslie Zwerwer

Version 0:

Reviewer comments:

Reviewer #1

(Remarks to the Author)

The manuscript by Zwerwer and colleagues is an interesting study exploring potential improvements of detection of SARS-CoV-2 infections in longitudinal studies, but also understanding the true rates of exposures to the virus. The study shows an innovative approach of using clustering of N IgG trajectories to identify infections. The current infection classification have several limitations that are (partly) overcome by the new approach of improved infection scoring and therefore are relevant to report. Compared to existing literature, the main addition to our current understanding of the COVID-19 pandemic is the approach to get an as accurate as possible estimation of the number of exposures/infections, which is helped by the large samples size.

The analyses are well-conducted and the manuscript is well written. However, the manuscript is quite technical, and the way of reporting (with many numbers and percentages) sometimes unclear and hard to follow. Also, the scientific and implications and applicability of the reported insights for understanding the COVID-19 pandemic or improved surveillance of future outbreaks are largely lacking from the manuscript.

General comments:

For a broader readership I would recommend to describe what infections are evaluated here. Given the manuscript mainly focusses on maximizing detection molecularly and serologically, the authors are working towards approaches of detecting any exposure to the virus, which includes exposures to the virus not leading to actual infections.

Paragraph starting with line 214: In this section, the authors use a fixed seropositivity threshold to identify infections. It is however not clear how this infection classification was performed. Was the person scored as infected at the first timepoint they crossed the seropositivity threshold? Or did every record of N IgG that was above the threshold count as an infection (allowing multiple infections within the follow up)?

A common method to detect infection in longitudinal designs is simply comparing levels pre and post and identify infection when an x-fold increase in antibody levels is observed. In the current manuscript the authors apply the manufacturers cutoff which is a useful comparison. However, I would be interested to see how the k-means clustering approach compares to fold-change approaches to detect infection.

Related to this, profiles being classified as flat, do show changes in levels, often falling back to previous levels. Are these very mild infections, or is this variability in the assay? And, could the authors briefly discuss how clustering deals with high versus low N antibody responses and whether this could impact identification of infections? The clustering overall works well for this dataset. Could the authors briefly discuss how well this approach may work for smaller datasets which are more commonly available to researchers?

In the introduction the authors briefly introduce the rationale for improving the detection of SARS-CoV-2 infections. I would welcome to, in light of this, elaborate on this in the discussion on potential implications for understanding the COVID-19 pandemic or potential future pandemics/

Specific and minor comments:

Line 93: what was the range of participant's ages? IQR is great to report in the suppl table as it is now, however, it does not give the full picture of age groups (e.g. it is not clear whether children were also included in the study).

Line 97: what was the reason for the lack of vaccination? Were they perhaps children? Or they left the study earlier (before receiving a vaccination?)

Line 103: is before study period corresponding with the very first visit / timepoint? Or was it self-reported? The first timepoint differed between participants.

Line 133: Remove “overall”

Line 133-135: “compared to 350 (0.2%) among those with ≥ 4 N-antibody measurements” – do you mean 350 out of the *total number* of those with ≥ 4 N antibody measurements? Or the ones that remained after excluding the abovementioned clusters (13 and 10)? Please Clarify.

Line 136: “final different trajectory-based classifications” – which classification was the “final”? was it the identity clustering or log2 clustering? Based on the Methods section, the log2 method was used for this particular visualization, but please specify it here as well.

Line 158-159: what do you mean by “inner 90% of all observations”? The geom_smooth (that according to Methods was used for this analysis) displays smoothed *mean* trajectories of all of the observations. Do you therefore mean that you have excluded the outliers falling outside the “inner 90%” (do you mean 90%CI? Or was it something connected with the clustering?)? Please specify / clarify (also clarify the procedure in the Methods).

Section within lines 160-173: I would suggest the authors to structure the paragraph slightly differently to make the message more clear. First, the authors may mention all the instances in which the results of the clustering analysis *agree with the swabs*. The authors may start with the overall agreement (just like it already is in the text): “Overall agreement between the N-antibody trajectories...” (sentence in lines 161-163), and then continue with the sentence describing in how many (and %) participants the *lack of infection* was scored by both swab and N-trajectories clustering, and next in how many the *infection* was scored by both methods (+ the part about the infection dates etc (lines 169-173). After describing all these “agreement” results, the authors may continue with describing the “disagreements” (starting with the sentence “28.6% of the 25,404 swab-positive...” (line 163-165). The authors may consider placing the “disagreement” results in a separate paragraph. This structure (first positives – agreement of both methods, and then disagreements – when the methods give different outcomes) may be easier to follow for the reader.

Line 191: What is then the estimated % of the total subpopulation that got infected?

Line 195-196: The results were “slightly different” in what way and compared to what? I would suggest to remove this first sentence of the paragraph - it is unnecessary and confusing. I would suggest the authors to start the paragraph with just: “When stratifying by vaccination status, 4.8-10.9% ... “ (“Overall” can be removed)

Line 200-201: Are these percentages consistent between virus variants?

Line 202-204 (first sentence of the paragraph): It is not clear to me what you compare in here? Please re-write for clarity.

Line 212: It is unclear what you report here. Are these the infections that you did not manage to detect by either methods? Or infections that were not detected by both methods together, but were detected by another method? The same for line 288-289

Line 265-270: I do not immediately see the importance of reporting the % of N-negative participants. What is the reason to include this additional analysis?

Line 292-293: the combination of swab data and serology data is frequently used in several ways; using swab data to validate assays, evaluation how well N seropositivity perform in a vaccination era or the detection of vaccine-breakthrough infections and generally identifying infections in population studies, however, usually not investigated this specifically and which these large numbers of subjects. I would suggest to adjust this statement accordingly.

Supplementary Figure 2: Box with Cluster14: Is the evidence of the previous infection (before the follow up, I assume?) based on serology, self-report via questionnaire, or the PCR of the swab?

(Remarks on code availability)
see comment to the editor

Reviewer #2

(Remarks to the Author)

(Remarks on code availability)

Reviewer #3

(Remarks to the Author)

This study examines the detection of SARS-CoV-2 infections that were previously undetected by conventional tests, such as nasal and throat swabs. This is a significant issue, as it is well-established that many cases go unnoticed, contributing to the silent spread of the virus. However, accurately determining the proportion of these silent cases remains challenging. The authors estimate that over 7% of true infections are missed by conventional tests and argue that routine monthly swabs are insufficient for identifying all positive cases.

The manuscript is clear and methodologically sound.

One minor critique is the reliance on nucleocapsid (N) antibody detection for the analysis. While this approach helps avoid confounding variables such as vaccination and cross-reactivity, it is known that not all infected individuals develop sufficient levels of N-antibodies. This limitation is particularly pronounced in individuals previously immunized with spike mRNA vaccines.

Overall, the manuscript is robust and makes a meaningful contribution to the field.

(Remarks on code availability)

Reviewer #4

(Remarks to the Author)

This study combines epidemiological data from a study of 270,686 participants in the UK, data on SARS-CoV-2 infection by PCR/antigen test, and data on measured antibody responses. The study appears to have been competently and professionally designed and implemented. Appropriate ethical approval is in place. Anti-Nucleocapsid antibodies were measured semi-quantitatively using a previously described research-use only assay. In terms of the scale of organisation, logistics, data management, and bio-banking, this is clearly an extraordinary study.

In late 2024, a great deal is known regarding the epidemiology of SARS-CoV-2. The novelty of this study is provided by the method for analysing anti-N antibody levels and inferring infections. A K-means algorithm was used to cluster anti-N antibody trajectories into 15 different groups, according to patterns of boosting and waning (Supp Figure 3). These 15 clusters are then aggregated to 6 higher order groups: "All < 10", "Flat", "Decreasing", "Increasing", "De-and increasing", "All > 200". The classification of these higher order groups is then compared to data on swab positivity (Figure 2). Encouragingly, there is good correspondence between categories, for example the "All < 10" and "Flat" groups often have no positive swabs. In my reading, this tells us that there are some infections that have been missed by swab testing but detected by anti-N antibodies, and conversely that there are some infections that have been inferred by anti-N antibodies, but not detected by swab testing. This makes a lot of sense to me. However, the novel quantitative result that is highlighted in the abstract is:

"After combining N-antibody (hypothetical) infections with swab-positivity, we estimated that 7.4% of all true infections would have remained undetected, 25.8% by swab-positivity-only and 28.6% by trajectory-based N antibody classifications only"

It's difficult to understand the significance of this statement.

Minor comments

In the abstract, please include confidence intervals with estimated quantities.

A classic epidemiological Table 1 would be very beneficial – something like either Supp Table 1 or Supp Table 2.

The colour coding of Figure 2 is not especially informative. It just represents the quantity of the data. It may be more informative to colour by row sums to demonstrate the concordance of the data in the style of a contingency table.

I wasn't able to find any information on the sensitivity and specificity of the anti-N antibody assay. This would have important consequences on the interpretation of the results. Furthermore, it's important to consider that sensitivity will decline with time since infection.

Figure 3b – do the two lines overlap?

How are multiple swab positives in one participant accounted for?

Optional comment

I note this comment as optional, because it's along the lines of what I would have done – as such the authors can safely ignore this comment. I find the application of an unsupervised learning/clustering algorithm to such a large dataset unnecessarily complicated. At its core, this is a simple problem – if someone gets infected, their anti-N antibodies boost up. The greater the boost, the greater the likelihood of infection. Resorting to such a complex clustering algorithm potentially risks obscuring the real pattern (boosting of antibodies following infection) with other properties of the antibody trajectory (e.g. antibody levels before boosting).

(Remarks on code availability)

NA

Version 1:

Reviewer comments:

Reviewer #1

(Remarks to the Author)

I would like to thank the authors for their clear and extensive rebuttal. The answers provided were clear and convincing. The changes made improved to manuscript. I would like to complement the authors on the study and the extensive additional work performed for the revised manuscript.

(Remarks on code availability)

Reviewer #2

(Remarks to the Author)

(Remarks on code availability)

Reviewer #4

(Remarks to the Author)

I am satisfied with the authors response to my previous comments.

(Remarks on code availability)

Thank you very much for giving us the opportunity to respond to the reviewers' comments which has substantially improved the manuscript. We have carefully revised our work and highlighted all changes in our point-by-point answers below.

Please note that page and line numbers refer to the tracked changes version with changes "in line" (not in balloons).

Reviewer #1 (Remarks to the Author):

1. The manuscript by Zwerwer and colleagues is an interesting study exploring potential improvements of detection of SARS-CoV-2 infections in longitudinal studies, but also understanding the true rates of exposures to the virus. The study shows an innovative approach of using clustering of N IgG trajectories to identify infections. The current infection classification have several limitations that are (partly) overcome by the new approach of improved infection scoring and therefore are relevant to report. Compared to existing literature, the main addition to our current understanding of the COVID-19 pandemic is the approach to get an as accurate as possible estimation of the number of exposures/infections, which is helped by the large samples size. The analyses are well-conducted and the manuscript is well written. However, the manuscript is quite technical, and the way of reporting (with many numbers and percentages) sometimes unclear and hard to follow. Also, the scientific and implications and applicability of the reported insights for understanding the COVID-19 pandemic or improved surveillance of future outbreaks are largely lacking from the manuscript.

- a. We thank the reviewer for the extensive, positive, and useful feedback on our manuscript and we highly appreciate the time taken for this thorough review. To improve the clarity of the manuscript we have made the following adjustments:

- We amended the manuscript based on the reviewer's general comments and line-by-line suggestions provided below, including restructuring paragraphs to improve overall readability.
- We have included a table (**Table 2**) in the revised manuscript with the estimates of infection and removed all confidence intervals from the relevant paragraphs to improve readability. (P. 11–12, L. 206–225 & P 15-17, L. 237-280)
- After doing this, we have carefully reviewed the manuscript and provided extra clarifications. For instance, in the paragraphs in which we discuss the estimated the number of true infections (Results):
 1. "Assuming that both types reflected true infections and there were no false-positives, using a method dependent capture-recapture model we estimated the true total number of infections during the study period among all participants with ≥ 4 N-antibody measurements (i.e. those detected and undetected with either N-antibody trajectory-based classifications or swab-positivity) as 34,249." (P. 12, L. 213 – 218)
 2. "As a sensitivity analysis, we reclassified the 505 participants with ≥ 60 days between the N-antibody (hypothetical) infection date and closest swab-positive infection and estimated the percentage of undetected infections with either N-antibody-based classifications or swab-positivity." (P. 15, L. 237 – 240)

- b. Additionally, in the revised manuscript we have included the scientific implications of our findings for understanding the COVID-19 pandemic and surveillance of future outbreaks (see response to comment 8).
 2. General comments: For a broader readership I would recommend to describe what infections are evaluated here. Given the manuscript mainly focusses on maximizing detection molecularly and serologically, the authors are working towards approaches of detecting any exposure to the virus, which includes exposures to the virus not leading to actual infections.
 - a. We are not completely sure of the reviewer's meaning here; we interpret their comment as referring to asymptomatic "infections" which some would argue are not "infections" in the traditional sense of having symptoms of infection. The challenge is that many asymptomatic infections have been shown to have high viral loads, and lead to onwards transmission, as well as providing protection from future infection. Hence for the purposes of surveillance and modelling, they have substantial importance (particularly since their asymptomatic nature may make onward transmission more likely as behaviour will not be modified). In this analysis, we are not considering exposure to the virus which does not lead to viral replication of a sufficient degree to be detectable by positive PCR or lead to an immunological response. In the revised version of our manuscript, we have clarified this as follows:

Introduction (last paragraph):

 - "We define SARS-CoV-2 infections as symptomatic or asymptomatic, but with sufficient replicating virus to be detectable on PCR or lead to seroconversion through an immunological response, i.e. sufficient replicating virus that onwards transmission could be possible." (P. 4 L. 92 – 95)
3. Paragraph starting with line 214: In this section, the authors use a fixed seropositivity threshold to identify infections. It is however not clear how this infection classification was performed. Was the person scored as infected at the first timepoint they crossed the seropositivity threshold? Or did every record of N IgG that was above the threshold count as an infection (allowing multiple infections within the follow up)?
 - a. We have clarified in the Methods section (P. 33 L. 671 – 680) that we used the fixed seropositivity threshold as follows:
 - "Secondly, we classified N-antibody trajectories using the manufacturer's proposed N-antibody seropositivity threshold of 30 ng/mL²¹. We considered a participant infected if the manufacturer's threshold of 30 ng/mL was crossed (upwards) during their study period. In all other scenarios, we considered a participant not infected during their study period, which could either mean they had an infection prior to their study period (i.e. first measurement ≥ 30 with no subsequent rise from < 30 to ≥ 30 ng/mL) or their N-antibodies measurements showed no evidence of an infection at all (all measurements < 30 ng/mL). Consistent with the trajectory-based analysis, and given the small number of participants with multiple swab-positives (350, 0.2%), we did not try to identify multiple N-antibody (hypothetical) infections during the study period."
4. A common method to detect infection in longitudinal designs is simply comparing levels pre and post and identify infection when an x-fold increase in antibody levels is

observed. In the current manuscript the authors apply the manufacturers cutoff- which is a useful comparison. However, I would be interested to see how the k-means clustering approach compares to fold-change approaches to detect infection.

- a. We thank the reviewer for providing this suggestion which we have incorporated in the revised manuscript, using a fourfold increase as a seropositivity criteria following ¹. In the main analysis we treated values below or above the limits of detection (10 and 200 ng/mL respectively) as equal to these limits; in a sensitivity analysis we treated antibody levels rising from below to ≥ 200 ng/mL as infected (since otherwise no value starting above 50 ng/mL could ever be considered an infection, regardless of the rise). For both fourfold N-antibody classifications we performed a capture-recapture analysis to estimate the number of undetected infections (P. 16, L. 265 – 272 & P. 16-17, L. 273 – 280), and summarised the results of all the capture-recapture analyses in a new Table 2. Additionally, we compared both fourfold-based classifications to the trajectory-based classifications using different data sources to define swab-positive infections (P. 17-19, L. 295 – 335, Figure 3 and P. 20, L. 349 – 365) and added text to the Discussion and Methods. Specifically:

Results:

- “ Table 2. Estimated number of true infections using different definitions of N-antibody seropositivity.” (P. 13-14, 226–228)
- “ When defining N-antibody (hypothetical) infections based on an arbitrary fourfold rise in consecutive antibody levels²² (treating values above or below the limits of detection as equal to these limits, see **Methods**), there were 30,047 (hypothetical) N-antibody or swab-positive infections (**Table 2**). 20,999 (69.9%) were identified using the fourfold criterion and 25,404 (84.5%) using swab-positivity; 16,356 (54.4%) were detected by both. Hence, under the assumption of no false positives as above, using a method dependent capture-recapture model, we estimated that 7.9% of a total of 32,615 true infections would have remained undetected by both swab-positivity and the fourfold N-antibody seropositivity criterion.” (P. 16, L. 265–272)
- “Finally, in a sensitivity analysis, we treated values rising to ≥ 200 ng/mL as automatically meeting the fourfold criteria, since in the main analysis treating these as ≥ 200 ng/mL antibody levels above 50 ng/mL could never rise to levels classified as infected. In this sensitivity analysis we identified 31,577 (hypothetical) infections (**Table 2**) of which 23,161 (73.3%) were identified by N-antibody and 25,404 (80.5%) by swab-positivity; 16,988 (53.8%) were identified by both. Hence, assuming no false positives (as above), we estimated a total of 34,634 true infections of which 8.8% would remained undetected by both methods.” (P. 16-17, L. 273–280)
- “ Next, we compared positivity based on N-antibody trajectories, the fixed 30 ng/mL threshold, and both the fourfold increases in N-antibody levels using different data sources to define swab-positive infections. Across the different data sources, the percentage of participants with N-antibody (hypothetical) infections that were identified using swab-

¹ Long, Q.-X. et al. Antibody responses to SARS-CoV-2 in patients with COVID-19. Nature Medicine 26, 845–848 (2020).

positivity ranged between 29.6-78.8% for the trajectory-based classification, between 22.4-60.5% for the fixed 30 ng/mL threshold, between 30.9-82.7% for the main fourfold-based classification and between 29.2–78.0% for the fourfold-based sensitivity analysis classification (**Fig. 3a**).” (P. 17-18, L. 297–304, Figure 3a)

- “The differences between both fourfold-based classifications and the trajectory-based classifications were much smaller; the main fourfold-based classification identified slightly fewer N-antibody (hypothetical) infections overall (**Table 2**) but a slightly higher proportion of these were swab-positive across all data sources (**Fig. 3a**).” (P. 18, L. 321 – 324)
- “The percentage of all N-antibody negative participants with no positive swab decreased from 99.0% to 94.7% with increasing richness of data source for the trajectory-based N-antibody classification, from 99.0% to 94.8 % for the threshold-based classification, from 98.6% to 93.7% for the fourfold-based N-antibody classification and from 98.7% to 94.1% for the fourfold-based sensitivity analysis (**Fig. 3b**).” (P. 19, L. 328– 332)
- “ In contrast, comparing main trajectory-based and alternative fourfold-rise-based classifications, smaller numbers of participants had positive fourfold-based and negative trajectory-based N-antibody classification (main N=281, **Supplementary Fig. 11**; sensitivity analysis N= 1,810, **Supplementary Fig. 12**). However, most of these (hypothetical) infections was swab-negative (68.0% and 84.3%, respectively), with generally decreasing N-antibody trends before the (hypothetical) infection, consistently with either a single falsely low measurement or prior infection. In contrast, the larger number of negative fourfold-based and positive trajectory-based N-antibody classification showed an evident increase in N-antibody levels in both main and sensitivity analysis (**Supplementary Fig. 11 & 12**). These N-antibody increases were either slightly blunted or started from higher initial levels and so did not reach a fourfold increase. Only 50.0% and 46.1% of these participants respectively were swab-positive during their study period.” (P. 20, L. 349–365)

Discussion:

- “ Differences between trajectory-based and fourfold-rise-based N-antibody classifications were much smaller (**Fig. 3, Table 2**). While only 50.0% of the trajectory-based (hypothetical) N-antibody infections and fourfold-rise-based seronegatives were swab-positive, the N-antibody measurements showed a clear increase in N-antibody levels (**Supplementary Fig. 11**), with N-antibodies starting from higher levels or responses being slightly blunted. Hence, while the fourfold-based N-antibody classifications were slightly better aligned with swab-positivity, trajectory-based classifications could offer advantages when blunted N-antibody responses might be expected (e.g. in vaccinated populations) or when N-antibody levels are higher (e.g. in previously infected populations), and are less susceptible to single erroneous measurements. Furthermore, whilst based on a previous study²², the fourfold rise was an arbitrary threshold.” (P. 24, L. 457 – 468)

- “Whilst N-antibodies serosurveillance can be used to improve estimates of the number of previous infections, for optimal use in large datasets, fourfold-based analysis or ideally trajectory-based analysis is required over threshold-based analysis.” (P. 27, L. 541–543)

Methods:

- “Thirdly, at the suggestion of a reviewer, in a sensitivity analysis we considered participants infected during the study period if their N-antibody levels rose fourfold over consecutive measurements²², treating values below or after the limits of detection (10 and 200 ng/mL respectively) as equal to these limits. Lastly, we performed a sensitivity analysis on these fourfold-based classifications, treating antibody levels rising from below to ≥ 200 ng/mL as infected (since otherwise no value starting above 50 ng/mL could ever be considered an infection, regardless of the rise). Consistent with the trajectory-based analysis, we did not try to identify multiple N-antibody (hypothetical) infections in both fourfold-based analyses.” (P. 33-34, L. 680 – 688)
- “Sensitivity analysis used classifications based on the manufacturer’s threshold, and both fourfold-based N-antibody classifications.” (P. 36, L. 735–736)

5. Related to this, profiles being classified as flat, do show changes in levels, often falling back to previous levels. Are these very mild infections, or is this variability in the assay?
 - a. There are several possible reasons why antibody trajectories classified as *flat* might show small changes in antibody levels. Firstly, the centering of the antibody trajectories on the midpoint between the maximum difference between any two consecutive measurements per participant, necessary to compare trajectories between infection groups fairly, can create a small but arbitrary distortion in those without swab-positive infections and classified as *flat* or *decreasing* which would not be apparent were we to instead label the first measurement as time 0 in this group. Secondly, changes in antibody levels can be caused by variability in the assay, which may be more pronounced in unspiked serum/plasma². Thirdly, infections could have occurred before the participant’s study period. Lastly, changes in antibody trajectories classified as *flat* might be caused by very mild infections (i.e. extremely blunted responses) or infections in immunocompromised participants (see also in Figure 1 the *flat* trajectories with a positive swab *during/before and during* the participant’s survey). In the revised manuscript we have addressed this as follows:

Discussion:

- “ While overall the clustering led to sensible infection groups, several antibody trajectories classified as *flat* by the trajectory-based N-antibody classification exhibited slight variations in antibody levels,

² Donaldson M, McBride J. Performance Evaluation Report for the N antibody assay used for research purposes in the protocol. (2021). Available at: <https://www.ndm.ox.ac.uk/covid-19/covid-19-infection-survey/n-antibody-assay-performance>

plausibly due to the centering of the antibody trajectories creating small but arbitrary distortions. Other explanations include variability in the assay, which is more pronounced in unspiked serum/plasma²¹, prior infections before the participant's study period and mild infection episodes or infection episodes in participants with compromised immune systems. Nevertheless, for those participants without swab-positives it remains uncertain if these variations were caused by such external influences or actual infections." (P. 25-26, L. 488 – 495)

6. And, could the authors briefly discuss how clustering deals with high versus low N antibody responses and whether this could impact identification of infections?
 - a. The clustering is affected by the complete shape of the antibody trajectory, which includes creating different clusters for high versus low N-antibody responses. In the revised version of Supplementary File 1 we have added further information on how both the shape and the height of the N-antibody response are incorporated into the clustering algorithm. Additionally, supplementary Figure 3 clearly visualizes how these differences are taken into account. For instance, clusters 1-3 from the id clustering were all classified as flat according to the trajectory-based N-antibody classification and have increasing heights. Various heights can also be found in the id clusters 4-8 (decreasing), id clusters 9-12 (increasing) and log clusters 1-5 (flat), log clusters 6-8 (decreasing) and log clusters 9-12 (increasing). The distinction between the different heights in antibody responses allows for careful analysis and classification of the clusters. We have made the following adjustments to our manuscript:

Methods:

- "This clustering method takes into account the shape and the height of the antibody trajectory. Hence, by considering both these characteristics the clustering can distinguish different clusters with similar shapes but different heights." (P. 30 L. 604 - 606)

Supplementary File 1:

- "Subsequently, each trajectory is assigned to a cluster based on the minimal dynamic time warping distance to each cluster centroid. Next to differences in shape, this distance considers differences in height."

7. The clustering overall works well for this dataset. Could the authors briefly discuss how well this approach may work for smaller datasets which are more commonly available to researchers?
 - a. A longitudinal variation of K-means can be applied to smaller datasets as well. We have added the following text to the Discussion in the revised manuscript to clarify important limitations:
 - " The longitudinal variation of K-means applied here could also be evaluated for other antibodies/infections to assess whether surveillance of immunity and infection incidence could be improved, but its application to smaller studies has several limitations. Firstly, each participant should have sufficient measurements to detect infections (e.g. here ≥ 4 measurements). Moreover, K-means can be sensitive to outliers, which can have large influence in smaller

datasets⁴¹, leading to less robust clustering or possibly overfitting. For smaller datasets, other clustering algorithms less sensitive to outliers, such as K-medoids, can be more appropriate⁴¹.” (P. 27, L. 519–526)

8. In the introduction the authors briefly introduce the rationale for improving the detection of SARS-CoV-2 infections. I would welcome to, in light of this, elaborate on this in the discussion on potential implications for understanding the COVID-19 pandemic or potential future pandemics

a. Our analyses showed that the use of N-antibody trajectories to define N-antibody infections has important advantages over the use of an arbitrary fixed threshold or a x-fold increase to define N-antibody infections. These methodological recommendations on serological detection could substantially improve estimates of infection. Moreover, after estimating the true number of infections using these serological positives together with swab-positives, we concluded that many SARS-CoV-2 infections remained undetected, especially when solely using swab-positives or N-antibody trajectory-based positives. These undetected infections complicate pandemic control. Furthermore, when not using these improved estimates of infection, the estimation of other parameters such as the infection fatality ratio will be biased. Such parameters, when used in subsequent studies simulating the pandemic or cost-effectiveness analyses, can lead to implementation of sub-optimal or even harmful interventions.

Based on above we have added and amended the following paragraphs in the Discussion of our revised manuscript.

- “Hence, while the fourfold-based N-antibody classifications were slightly better aligned with swab-positivity, trajectory-based classifications could offer advantages when blunted N-antibody responses might be expected (e.g. in vaccinated populations) or when N-antibody levels are higher (e.g. in previously infected populations), and are less susceptible to single erroneous measurements. Furthermore, whilst based on a previous study²², the fourfold rise was an arbitrary threshold.” (P. 24, 462–468)
- “The main study strength is our use of a longitudinal variation of K-means to identify infections from N-antibody trajectories. The challenge of using an arbitrary fixed threshold or a x-fold increase is that the boost in antibodies following infection is not consistent between individuals; for different reasons, some individuals will have a larger or smaller increase, from a higher or lower initial starting value. Both types of heterogeneity cause problems for simple definitions based on fixed or relative thresholds, problems which are magnified by censoring by limits of detection (here 10 and 200 ng/mL). N-antibodies were generally assayed monthly and comparing how antibody levels changed over time allowed us to still classify participants with “blunted” responses as having been infected and also taking into account declines over time which could affect sensitivity of fixed or relative thresholds.” (P. 25, 469–479)
- “The current study showed that estimates of SARS-COV-2 infection based on PCR- and LFT surveillance or serosurveillance greatly underestimate the true burden of infection. Even combining these methods did not provide a complete overview of infection. These underestimations could bias the estimation of other parameters, such

as the infection fatality ratio⁴². The results of subsequent studies simulating the pandemic or cost-effectiveness analyses may then become influenced by these biased parameters, which can lead to implementation of interventions that appear to be optimal in modelling but that lead to overall population harm when implemented in practice. Moreover, our study underlines that many SARS-CoV-2 infections remained undetected. While contributing to virus spread, these undetected infections might substantially complicate pandemic control. Our findings suggest that to optimize pandemic surveillance a combination of serosurveillance with swab-positivity should be used.” (P. 27, L. 527 – 538)

- “Whilst N-antibodies serosurveillance can be used to improve estimates of the number of previous infections, for optimal use in large datasets, fourfold-based analysis or ideally trajectory-based analysis is required over threshold-based analysis.” (P. 27, L. 541–543)

Specific and minor comments:

9. Line 93: what was the range of participant’s ages? IQR is great to report in the suppl table as it is now, however, it does not give the full picture of age groups (e.g. it is not clear whether children were also included in the study).
 - a. Due to privacy restrictions and possible disclosure we are not allowed to provide the age range in this study. However, in the revised version of our manuscript we have provided the 1st and 99th percentile of age in both Table 1 and Supplementary Table 1. The sample did include some children, with ethical approval to approach children down to 8 years for fingerprick sampling, although response was relatively low, leading to the 1st and 99th percentile of age in the full sample being [15, 85]. In the sample with respectively <4 and ≥4 measurements the 1st and 99th percentile were [10, 86] and [18, 84] respectively.
10. Line 97: what was the reason for the lack of vaccination? Were they perhaps children? Or they left the study earlier (before receiving a vaccination?)
 - a. Reasons for remaining unvaccinated were not collected, but we have clarified in the revised manuscript results section that:
 - “Respectively, 7.3%, 28.4%, 58.5% and 0.1% of participants had received 1,2, 3 and 4 vaccinations by the end of the period in which they had N-antibodies measured (denoted their study period), with 5.7% participants remaining unvaccinated throughout (e.g. due to age, ending study participation or personal choice).” (P. 5, L. 104 – 108)
11. Line 103: is before study period corresponding with the very first visit / timepoint? Or was it self-reported? The first timepoint differed between participants.
 - a. As described in the original Methods section (P. 29, 585 – 586), in this manuscript the participant’s study period refers to the period in which participants had N-antibodies measured and differs therefore per participant. Hence, participants with a swab-positive infection before the study period only, were swab-positive before their first N-antibody measurement was taken. In our manuscript we described this as follows:

Methods:

- “ The study period was defined as the period in which participants had N-antibody measurements available.” (P. 29, 585 – 586)

12. Line 133: Remove “overall”

- a. Removed as suggested.

13. Line 133-135: “compared to 350 (0.2%) among those with ≥ 4 N-antibody measurements” – do you mean 350 out of the *total number* of those with ≥ 4 N-antibody measurements? Or the ones that remained after excluding the abovementioned clusters (13 and 10)? Please Clarify.

- a. To clarify, we meant 350 (0.2%) out of the total number of participants with ≥ 4 N-antibody measurements. We have amended the revised manuscript as follows:

- “20 (37.0%) of these participants had two or more swab-positive infections during their study period (compared to 350 (0.2%) among all participants with ≥ 4 N-antibody measurements).” (P. 7-8, L. 146 – 148)

14. Line 136: “final different trajectory-based classifications” – which classification was the “final”? was it the identity clustering or log₂ clustering? Based on the Methods section, the log₂ method was used for this particular visualization, but please specify it here as well.

- a. As described in the original Methods, the final trajectory-based classification was based on the consensus of the id and log₂ N-antibody classifications, with manual reconciliation where these disagreed (**Supplementary Figures 2, 6 and 7**). We apologise that this may have been unclear in the original submission due to an incorrect reference to the figures in the methods section. In the revised manuscript we have corrected these references, and made further amendments, specifically:

Results:

- “A final trajectory-based classification was obtained based on consensus: where the two transformations differed (N=9,644, 9.7%), often relating to smaller absolute increases which were magnified on the relative (log) scale, participants were classified using visualization of the trajectories (**Supplementary Fig. 6&7**).” (P. 7, L. 139 – 143)

Methods:

- “We arbitrarily classified the N-antibody trajectories based on expected trajectories following infection (**Supplementary Fig. 3**) and then took the consensus of the id and log₂ N-antibody classifications, with manual reconciliation where these disagreed (see **Supplementary Fig. 2, 6&7 and Results**), and compared the combined final classification with swab-positive infections as defined above (**Fig. 1&2**).” (P. 31, L. 623 – 628)

15. Line 158-159: what do you mean by “inner 90% of all observations”? The geom_smooth (that according to Methods was used for this analysis) displays smoothed *mean* trajectories of all of the observations. Do you therefore mean that you have excluded the outliers falling outside the “inner 90%” (do you mean 90%CI?)

Or was it something connected with the clustering?)? Please specify / clarify (also clarify the procedure in the Methods).

- a. The reviewer is correct that to create the smooths in the figures we have excluded all outliers in terms of time (in days) falling below or above the 10th and 90th percentile – these exclusions are only for visualization of “average” trajectories and are not part of any analysis. Hence, after centering the days of the N-antibody measurements on the midpoint of the maximum increase in N-antibody levels, we applied the `geom_smooth(method='gam')` function to all N-antibody measurements assayed between the 10th and 90th percentile of the centered days. In the revised version of our manuscript we have described this as follows:

Fig. caption 1:

- “Black line depicts a generalised additive modelling smooth for all N-antibody measurements assayed between the 10th and 90th percentile of the centered days in each cluster.” (P. 42, L. 897 – 899)
- Additionally, we have amended all figure captions referring to the smooth in the Supplementary files accordingly (see **Supplementary Figures 3, 6, 7, 10 – 12**).

Methods:

- “We then visualised the N-antibody trajectories in each cluster together with a generalised additive model smooth (function `'geom_smooth(method = 'gam')` from `ggplot2`⁴⁸). The smoothing function was applied to all days (centered on the midpoint of the maximum increase in N-antibody levels) falling between the 10th and 90th percentile.” (P. 31, L. 619 – 623)

16. Section within lines 160-173: I would suggest the authors to structure the paragraph slightly differently to make the message more clear. First, the authors may mention all the instances in which the results of the clustering analysis *agree with the swabs*. The authors may start with the overall agreement (just like it already is in the text): “Overall agreement between the N-antibody trajectories...” (sentence in lines 161-163), and then continue with the sentence describing in how many (and %) participants the *lack of infection* was scored by both swab and N-trajectories clustering, and next in how many the *infection* was scored by both methods (+ the part about the infection dates etc (lines 169-173). After describing all these “agreement” results, the authors may continue with describing the “disagreements” (starting with the sentence “28.6% of the 25,404 swab-positive...” (line 163-165). The authors may consider placing the “disagreement” results in a separate paragraph. This structure (first positives – agreement of both methods, and then disagreements – when the methods give different outcomes) may be easier to follow for the reader.

- a. We have revised this paragraph as suggested by the reviewer (see P. 10, L. 174 – 195), specifically starting with all instances where the N-antibody trajectory-based analysis agreed with the swab-positivity results (i.e. total agreement, lack of infection, infection before the study period, infection during the study period). After this we discuss the estimated N-antibody (hypothetical) infection dates and we end this section with the disagreements as suggested. While we appreciate the reviewer’s suggestion to put the disagreement results in a separate paragraph, we believe that keeping these

sentences within the current paragraph improves the flow and readability, avoiding the creation of a very short paragraph.

17. Line 191: What is then the estimated % of the total subpopulation that got infected?
- a. Using N-antibody trajectories and swab-positivity we detected 31,716 infections during the study period in 31,364 participants. Using a method-dependent capture-recapture model we estimated that there would have been 34,249 true infections during the study period among all participants with ≥ 4 N-antibody measurements (**Table 2**), so an extra 2533 infections. When assuming that the missed infections were singletons and were not present in individuals with a detected infection during the study period, this would mean that $31,364 + 2,533 = 33,897$ of those with ≥ 4 N-antibody measurements were infected, or 18.3% of this population. We have incorporated this information into the revised version of our manuscript:
 - “Using both N-antibody trajectory-based classifications and swab-positive infections (including multiple swab-positive infections per participant), we identified 31,716 infections during the study period in 31,364/185,646 (16.9%) participants with ≥ 4 N-antibody measurements (**Table 2**).” (P. 11–12 , L. 207–210)
 - “Assuming that both types reflected true infections and there were no false-positives, using a method dependent capture-recapture model we estimated the true total number of infections during the study period among all participants with ≥ 4 N-antibody measurements (i.e. those detected and undetected with either N-antibody trajectory-based classifications or swab-positivity) as 34,249. Of these infections 7.4% remained undetected with either method, 25.8% by swab-positivity and 28.6% by N-antibody trajectory-based classification. Hence, assuming missed infections were singletons and that they only occurred for participants without an infection detected by either N-antibody trajectory-based analysis or swab-positives, 18.3% of participants with ≥ 4 N-antibody measurements would have been infected during the study period.” (P. 12, L. 213–225)
18. Line 195-196: The results were “slightly different” in what way and compared to what? I would suggest to remove this first sentence of the paragraph - it is unnecessary and confusing. I would suggest the authors to start the paragraph with just: “When stratifying by vaccination status, 4.8-10.9% ... “ (“Overall” can be removed)
- a. We have removed the first sentence of this section as well as the word Overall as suggested.
19. Line 200-201: Are these percentages consistent between virus variants?
- a. Thank you for raising this point. The 95% CI around the estimates of the percentage of true infections that were undetected do not overlap, so whilst it is not possible for us to directly test this within a single capture recapture model (rather separate models were fitted to each subgroups), the data support the percentages being different. We have added some possible reasons for the differences to the Discussion, as follows:
 - “Further, we found that estimates of undetected infections varied substantially across virus variants, with considerably more undetected infections during the Alpha epoch. During the Alpha epoch only one third of all detected infections was ascertained by swab-positivity (vs. 76.8% and 88.4% during the Delta and BA.1 epoch, see

Supplementary Table 4). These differences may have been caused by the limited availability of testing during the Alpha epoch (i.e. in the UK LFT were made widely available in April 2021, at the end of the Alpha epoch²⁵). Earlier research also showed that estimates of undetected infections vary by age group, variant, and region and that these variations may be related to differences in symptoms/disease severity, public sentiment and availability of testing²³.” (P. 22, L. 396 – 405)

20. Line 202-204 (first sentence of the paragraph): It is not clear to me what you compare in here? Please re-write for clarity.

a. We have rewritten this sentence as follows:

- “As a sensitivity analysis, we reclassified the 505 participants with ≥ 60 days between the N-antibody (hypothetical) infection date and closest swab-positive infection and estimated the percentage of undetected infections with either N-antibody-based classifications or swab-positivity.” (P. 15, 237 – 240)

21. Line 212: It is unclear what you report here. Are these the infections that you did not manage to detect by either methods? Or infections that were not detected by both methods together, but were detected by another method? The same for line 288-289

a. Indeed, these lines report the percentage of infections that remained undetected by either method. These infections were not detected by any other method. Instead, we estimated this number of undetected infections using a capture-recapture model. As described in the original Methods section, a capture-recapture model fits a loglinear model to the number of infections identified by swabs, N-antibody trajectories and their intersection to estimate the number of infections missed by either method. We have estimated the number of infections undetected by either swab-positivity and infections identified from the N-antibody trajectories (P. 11-12, L. 207 – 225). Additionally, we have performed several sensitivity and subgroup analyses (P. 15–17, L. 229 – 280, **Table 2, Supplementary Table 4**), including reclassifying participants with ≥ 60 days between the N-antibody (hypothetical) infection date and closest swab-positive infection (P. 15 L. 237 – 250), using the manufacturer’s threshold to define N-antibody infections (P. 15-16, L. 251 – 264), and using two different fourfold criteria to define N-antibody infections (P. 16, L. 265 – 272 & P. 16-17, L. 273 – 280). To clarify these analyses, we have carefully revised these paragraphs in the Results section and added a new summary Table 2. For instance:

Main analysis:

- “ Using both N-antibody trajectory-based classifications and swab-positive infections (including multiple swab-positive infections per participant), we identified 31,716 infections during the study period in 31,364/185,646 (16.9%) participants with ≥ 4 N-antibody measurements (**Table 2**). 24,440 (77.1%) of these detected infections were identified using N-antibody trajectory-based analysis, 25,404 (80.1%) were detected with swab-positivity and 18,128 (57.2%) were detected with both swab-positivity and N-antibody trajectory-based analysis. Assuming that both types reflected true infections and there were no false-positives, using a method dependent capture-recapture

model we estimated the true total number of infections during the study period among all participants with ≥ 4 N-antibody measurements (i.e. those detected and undetected with either N-antibody trajectory-based classifications or swab-positivity) as 34,249. Of these infections 7.4% remained undetected with either method, 25.8% by swab-positivity and 28.6% by N-antibody trajectory-based classification. Hence, assuming missed infections were singletons and that they only occurred for participants without an infection detected by either N-antibody trajectory-based analysis or swab-positives, 18.3% of participants with ≥ 4 N-antibody measurements would have been infected during the study period.” (P. 11–12, L. 207 – 225)

First sensitivity analysis:

- “As a sensitivity analysis, we reclassified the 505 participants with ≥ 60 days between the N-antibody (hypothetical) infection date and closest swab-positive infection and estimated the percentage of undetected infections with either N-antibody-based classifications or swab-positivity. Where the swab-positive infection date was ≥ 60 days before the N-antibody (hypothetical) infection date, we classified the infection as detected by swab-positivity only, and as N-antibody only when the swab-positive infection date was ≥ 60 days after the N-antibody (hypothetical) infection date. Under these assumptions, of all detected infections, 24,139 (76.1%) were detected using N-antibody trajectory-based analysis, 25,200 (79.5%) using swab-positivity and 17,623 (55.6%) by both methods (**Table 2**). Under the assumption that neither method identifies any false positives, using a method dependent capture-recapture model we estimated a total of 34,517 true infections during the study period, 8.1% of which would have been undetected by both swab-positivity and trajectory-based N-antibody positivity.” (P. 15, L. 237–250)

Discussion:

- “We found that under the assumption that swab-positives and N-antibody positives both reflect true infections, 7.4% of all true SARS-CoV-2 infections (i.e. those detected and undetected by swab-positivity and N-antibody trajectory-based classifications) would have remained unidentified from both swab results and N-antibody trajectories (compared to 25.8% by swab-only and 28.6% by trajectory-based N-antibody classifications only).” (P. 21, L. 372 – 377)

22. Line 265-270: I do not immediately see the importance of reporting the % of N-negative participants. What is the reason to include this additional analysis?

- a. In an ideal world, every participant who was infected would be swab-positive and N-antibody positive, and every participant who was not infected would be swab-negative and N-antibody negative. These percentages therefore give the effective specificity of using N-antibody to define infection. We have clarified this in the revised manuscript as follows:

Methods:

- “ For each, we estimated the percentage of swab-positive infections among those with N-antibody (hypothetical) infections (considering only the closest swab-positive infection to the N-antibody (hypothetical) infection date) and the percentage of participants without swab-positive infections among those without N-antibody (hypothetical) infections (N-antibody trajectory-based classifications: *all* ≤ 10 , *flat*, *decreasing* or *all* ≥ 200). These percentages are equivalent to estimating the sensitivity and specificity of swab-positivity using N-antibody (hypothetical) infections as a reference.” (P. 35-36, L. 728 – 735)

23. Line 292-293: the combination of swab data and serology data is frequently used in several ways; using swab data to validate assays, evaluation how well N seropositivity perform in a vaccination era or the detection of vaccine-breakthrough infections and generally identifying infections in population studies, however, usually not investigated this specifically and which these large numbers of subjects. I would suggest to adjust this statement accordingly.

- a. As far as we are aware, the effectiveness of identifying infections using both methods together and thereby estimating the number infections that remained undetected by either method has not been evaluated before, especially not on this scale. To clarify this, we have revised this sentence as follows:

- “ As far as we are aware, no other study has examined the efficacy of the combination of swab-positivity and N-antibody serological testing to identify SARS-CoV-2 infections and used this combination to estimate the number of infections remaining undetected by either method, particularly not on this scale.” (P. 21, L. 378 – 381)

24. Supplementary Figure 2: Box with Cluster14: Is the evidence of the previous infection (before the follow up, I assume?) based on serology, self-report via questionnaire, or the PCR of the swab?

- a. In cluster 14 all N-antibody measurements were >200 ng/mL, suggesting evidence of an infection before the participant’s study period based on the N-antibody levels (i.e. N-antibody serology). We have revised Supplementary Figure 2 to reflect this.

Reviewer #1 (Remarks on code availability):

see comment to the editor

Reviewer #2 (Remarks to the Author):

- a. Thank you for your contribution to the peer review of this manuscript.

Reviewer #3 (Remarks to the Author):

26. This study examines the detection of SARS-CoV-2 infections that were previously undetected by conventional tests, such as nasal and throat swabs. This is a significant issue, as it is well-established that many cases go unnoticed, contributing to the silent spread of the virus. However, accurately determining the proportion of these silent cases remains challenging. The authors estimate that over 7% of true infections are missed by conventional tests and argue that routine monthly swabs are insufficient for identifying all positive cases.

The manuscript is clear and methodologically sound.

a. We appreciate the time and effort taken for this careful review and value the constructive and helpful feedback on our manuscript.

27. One minor critique is the reliance on nucleocapsid (N) antibody detection for the analysis. While this approach helps avoid confounding variables such as vaccination and cross-reactivity, it is known that not all infected individuals develop sufficient levels of N-antibodies. This limitation is particularly pronounced in individuals previously immunized with spike mRNA vaccines.

Overall, the manuscript is robust and makes a meaningful contribution to the field.

a. We agree with the reviewer and note that their concern is the main reason for pursuing the trajectory-based analysis approach, rather than relying on a fixed threshold or relative increase – we have expanded our Discussion around this (P. 25, L. 469–479). We also found that N-antibody seroconversion was less likely among individuals who were more (recently or frequently) vaccinated, as they note. The limitations of using N-antibodies to identify infections are a key point of our manuscript. Since both N-antibody classifications and swab-positivity are unable to identify all infections separately, we examined the effectiveness of combining both approaches. Using a method dependent capture-recapture model we estimated the true total number of infections (i.e. those detected and undetected with either N-antibody trajectory-based classifications or swab-positivity), which enabled us to estimate the percentage of infections missed by either approach (P. 11–17, L. 207–280, **Table 2**). Additionally, to account for differences in seroconversion rates among vaccinated individuals we performed the same calculations on subgroups stratified by different vaccination status (P. 15, L. 229–236, **Supplementary Table 4**).

We hope this clarifies how we have already incorporated these limitations in our manuscript.

Reviewer #4 (Remarks to the Author):

28. This study combines epidemiological data from a study of 270,686 participants in the UK, data on SARS-CoV-2 infection by PCR/antigen test, and data on measured antibody responses. The study appears to have been competently and professionally designed and implemented. Appropriate ethical approval is in place. Anti-Nucleocapsid antibodies were measured semi-quantitatively using a previously described research-use only assay. In terms of the scale of organisation, logistics, data management, and bio-banking, this is clearly an extraordinary study.

In late 2024, a great deal is known regarding the epidemiology of SARS-CoV-2. The novelty of this study is provided by the method for analysing anti-N antibody levels and inferring infections. A K-means algorithm was used to cluster anti-N antibody trajectories into 15 different groups, according to patterns of boosting and waning

(Supp Figure 3). These 15 clusters are then aggregated to 6 higher order groups: “All < 10”, “Flat”, “Decreasing”, “Increasing”, “De-and increasing”, “All > 200”. The classification of these higher order groups is then compared to data on swab positivity (Figure 2). Encouragingly, there is good correspondence between categories, for example the “All < 10” and “Flat” groups often have no positive swabs. In my reading, this tells us that there are some infections that have been missed by swab testing but detected by anti-N antibodies, and conversely that there are some infections that have been inferred by anti-N antibodies, but not detected by swab testing. This makes a lot of sense to me.

However, the novel quantitative result that is highlighted in the abstract is:

“After combining N-antibody (hypothetical) infections with swab-positivity, we estimated that 7.4% of all true infections would have remained undetected, 25.8% by swab-positivity-only and 28.6% by trajectory-based N antibody classifications only”

It’s difficult to understand the significance of this statement.

- a. We highly value the time taken for this thorough review and would like to thank the reviewer for the in-depth comments and useful feedback on our manuscript.

In the Discussion, we have added some further detail on the impact of this statement ((P. 27, L. 527 – 538), see also response to Reviewer #1 point 8), but we are not able to provide further detail in the Abstract given the word count limits. We have reworded the Abstract sentence to try to make this clearer (P. 2, L. 37–41).

Minor comments

29. In the abstract, please include confidence intervals with estimated quantities.
 - a. We have included these as suggested:
“ Using N-antibody (hypothetical) infections and swab-positivity, we estimated that 7.4% (95%CI: 7.0–7.8%) of all true infections (detected and undetected) were undetected by both approaches, 25.8% (25.5–26.1%) by swab-positivity-only and 28.6% (28.4–28.9%) by trajectory-based N-antibody-classifications-only.” (P. 2, L. 37–41)
30. A classic epidemiological Table 1 would be very beneficial – something like either Supp Table 1 or Supp Table 2.
 - a. Based on this comment we have moved Supplementary Table 1 to the main text and amended all references to tables accordingly.
31. The colour coding of Figure 2 is not especially informative. It just represents the quantity of the data. It may be more informative to colour by row sums to demonstrate the concordance of the data in the style of a contingency table.
 - a. We have now color coded the cells by the row percentage instead of the quantity of the data as suggested.
32. I wasn’t able to find any information on the sensitivity and specificity of the anti-N antibody assay. This would have important consequences on the interpretation of the

results. Furthermore, it's important to consider that sensitivity will decline with time since infection.

- a. We had referred to the manufacturer's document in the original version, but in the revised manuscript we have now included the estimated sensitivity and specificity of the specific N-antibody assay. Additionally, the reviewer points correctly out that sensitivity will decline with time since infection. This was another reason for focusing mainly on the trajectory-based analysis of N-antibodies to define infections during the participant's study period in this study. We have revised our manuscript as follows:

Introduction

- “Nevertheless, the sensitivity of N-antibodies to detect infections depends strongly on the population, the time since infection and the thresholds used; previous studies have reported sensitivities ranging from 40-100%^{4, 12, 13, 14, 15, 16.}” (P. 3, L. 66-69)

Methods:

- “SARS-CoV-2 N-antibody levels were tested using a research-use only assay (details in²¹). At the manufacturer's threshold of 30 ng/mL, the sensitivity of this assay was 94.3% and specificity 92.8%.” (P. 29 L. 581 – 583)

Discussion:

- “N-antibodies were generally assayed monthly and comparing how antibody levels changed over time allowed us to still classify participants with “blunted” responses as having been infected and also taking into account declines over time which could affect sensitivity of fixed or relative thresholds.” (P. 25, L. 475 – 479)

33. Figure 3b – do the two lines overlap?

- a. Indeed, the lines in Figure 3b (largely) overlap. In the revised version of our manuscript there are now four lines (based on addition of two further analyses in response to reviewer 1 point 4). We have included a sentence to clarify this:
Caption Figure 3:
 - “ In Figure 3b the four lines (nearly) overlap.” (P. 48, L. 1038)

34. How are multiple swab positives in one participant accounted for?

- a. Firstly, we recognize that participants can test positive for extended periods of time when testing is independent of symptoms (such as in the COVID-19 infection survey). Therefore, as described in the original Methods, we incorporated information from genetic sequencing, S-gene presence/absence, and Ct values, together with negative PCR test results from the survey only to define different swab-positive infection episodes, grouping multiple swab-positives that were plausibly from the same infection together. Secondly, given the very small number of participants with evidence of multiple infection episodes based on swab-positivity during their study period (350 (0.2%)), we did not attempt to identify second and third infections from N-antibody data for any method. In theory this could have been possible for the N-antibody trajectory clustering approach, and in **Supplementary Figure 7** we note this

possibility for 54 participants in cluster 13 using identity clustering and cluster 10 using log2 transformed clustering. However, due to fluctuations around a fixed or relative threshold, defining multiple infections for other N-antibody methods would be very complex. We have clarified in the Methods that we did not attempt this and considered the very limited number of second and third swab-positive episodes during the participant's study period as infections detected only by swab-positivity.

Definition swab-positive episodes:

- “To reflect the fact that some individuals can test positive on PCR for extended periods of time when testing is independent of symptoms/case contacts as in the survey (in contrast to national testing programmes), whereas others have reinfections (confirmed by sequencing) after only short periods of time, we incorporated information from genetic sequencing, S-gene presence/absence, and Ct values, together with negative PCR test results from the survey only⁴⁴.” (P. 30, L. 593 – 598)

The capture-recapture analysis:

- “ For participants with multiple swab-positive infections, we considered the closest swab-positive infection to the N-antibody (hypothetical) infection date detected by both methods and all other swab-positive infections detected by swab-positivity only.” (P. 32, L. 652–655)
- “Consistent with the trajectory-based analysis, and given the small number of participants with multiple swab-positives (350, 0.2%), we did not try to identify multiple N-antibody (hypothetical) infections during the study period.” (P. 33, L. 677–680)
- “Consistent with the trajectory-based analysis, we did not try to identify multiple N-antibody (hypothetical) infections in both fourfold-based analyses.” (P. 33-34, L. 686–688)

The logistic regression:

- “For participants with *increasing/de- and increasing* N-antibody trajectories who had multiple swab-positive infections, we considered the closest swab-positive infection to the N-antibody (hypothetical) infection date a responder and all other swab-positive infections non-responders.” (P. 34, L. 703–706)

The analysis of different data sources:

- “For each, we estimated the percentage of swab-positive infections among those with N-antibody (hypothetical) infections (considering only the closest swab-positive infection to the N-antibody (hypothetical) infection date) and the percentage of participants without swab-positive infections among those without N-antibody (hypothetical) infections (N-antibody trajectory-based classifications: *all* ≤ 10 , *flat*, *decreasing* or *all* ≥ 200).” (P. 35, L. 728 – 733)

Optional comment

35. I note this comment as optional, because it's along the lines of what I would have done – as such the authors can safely ignore this comment. I find the application of

an unsupervised learning/clustering algorithm to such a large dataset unnecessarily complicated. At its core, this is a simple problem – if someone gets infected, their anti-N antibodies boost up. The greater the boost, the greater the likelihood of infection. Resorting to such a complex clustering algorithm potentially risks obscuring the real pattern (boosting of antibodies following infection) with other properties of the antibody trajectory (e.g. antibody levels before boosting).

- a. The challenge as described by reviewer #3 is that the boost in antibodies is not consistent between individuals – for different reasons some people will have a larger or smaller increase, from a higher or lower initial starting value. Both types of variation cause problems for simple definitions based on fixed or relative thresholds, problems which are magnified by censoring by limits of detection (here 10 and 200 ng/mL). We have expanded our discussion around this (P. 25, L. 469–479). The trajectory-based approach is a method to overcome these challenges and, as described in our manuscript, were much better aligned with swab-positivity compared to the fixed threshold-based classifications with substantial differences between the estimates of undetected infections. There were smaller differences with the relative threshold-based approaches now incorporated in response to reviewer 1 point 4 and added some further discussion around this (P. 24, L. 457–468).

Reviewer #4 (Remarks on code availability):

NA